# MuPT: A Generative Symbolic Music Pre-trained Transformer

**Xingwei Qu**[1 3 4*], **Yuelin Bai**[5*], **Yinghao Ma**[1 7*],
**Ziya Zhou**[3], **Ka Man Lo**[3], **Jiaheng Liu**[1 17], **Ruibin Yuan**[1 3], **Lejun Min**[8 16], **Xueling Liu**[1],
**Tianyu Zhang**[9], **Xinrun Du**[1], **Shuyue Guo**[1], **Yiming Liang**[10], **Yizhi Li**[1 4], **Shangda Wu**[11],
**Junting Zhou**[12], **Tianyu Zheng**[1 19], **Ziyang Ma**[13 18], **Fengze Han**[1], **Wei Xue**[3], **Gus Xia**[8],
**Emmanouil Benetos**[7], **Xiang Yue**[1], **Chenghua Lin**[4], **Xu Tan**[14], **Wenhao Huang**[19]
**Jie Fu**[3†], **Ge Zhang**[1 19* †]

[1]M-A-P, [2]University of Waterloo, [3]HKUST, [4]University of Manchester,
[5]Shenzhen Institute of Advanced Technology, CAS, [6]Vector Institue, [7]QMUL, [8]MBZUAI,
[9]Mila, [10]Institute of Automation, CAS, [11]Central Conservatory of Music,
[12]Institute for AI, [13]SJTU, [14]MSRA, [15] University of Montreal,
[16]CCRMA, Stanford University,[17] NJU,[18]Nanyang Technological University, [19]ByteDance

## Abstract

In this paper, we explore the application of Large Language Models (LLMs) to the pre-training on symbolic music. While the prevalent use of MIDI in music modeling is well-established, our findings suggest that LLMs are inherently more compatible with ABC Notation, which aligns more closely with their design and strengths, thereby enhancing the model's performance in musical composition. To address the challenges associated with misaligned measures from different tracks during generation, we propose the development of a Synchronized Multi-Track ABC Notation (**SMT-ABC Notation**), which aims to preserve coherence across multiple musical tracks. Our contributions include a series of models capable of handling up to 8192 tokens, covering 90% of the symbolic music data in our training set. Furthermore, we explore the implications of the Symbolic Music Scaling Law (**SMS Law**) on model performance. In music structure experiments on repetition, we outperform GPT-4 by 17% (average Intra Similarity and Repetition Rate) on the full test set, and surpass the SOTA ABC-notation model ChatMusician by 6% on the single-track test set. In terms of subject evaluation, listeners preferred music from our system in 79% of cases, comparing to GPT-4. The results indicate a promising direction for future research in music generation, offering extensive resources for community-led research through our open-source contributions.

## 1 Introduction

Large Language Models (LLMs) have experienced remarkable advancements, leading to their broad application across numerous domains. As these models extend into multimodal areas, such as visual and auditory fields, their capability to represent and model complex information, including images (Liu et al., 2023) and speech (Baevski et al., 2020) becomes increasingly critical. However, this expansion also highlights significant challenges that must be addressed. Specifically, the development of effective tokenizers for images and videos, as well as advanced codecs for the audio domain.

In the domain of music, LLMs encounter inherent challenges that hinder their effective utilization. These models often struggle to capture the consistency of long-term structural consistency of music essential for pleasing music (Dai et al., 2022; Briot & Pachet, 2020; Dai et al., 2021). This issue stems from the use of Musical Instrument Digital Interface (MIDI), which, while effective, poses significant challenges in terms of music's readability and structural representation. The widely-used performance MIDI data may lack structural annotations and cannot inherently encode phenomena such as music repetition, thus resulting in longer sequence lengths (Yuan et al., 2024).

To tackle this issue, the integration of ABC notation offers a novel approach to overcoming the limitations of MIDI formats, visualized in Figure 1. Yuan et al. (2024) advocate for this method, highlighting ABC notation's readability and structural coherence. Their methodology involves fine-tuning the LLAMA2 model, leveraging instruction tuning to enhance the model's musical output capabilities (Touvron et al., 2023b;a). The research overlooks critical tokenization considerations within musical compositions.

In this paper, we aim to propose a training standard with transformer decoder-only architecture for symbolic music generation tasks, which is suitable for single / multi-track music generation. We observe that mismatches between measures can occur by employing the traditional 'next-token-prediction' paradigm for symbolic data training. This issue arises because ABC notations are generally notated track by track, completing one track before moving on to the next. To address this challenge, we propose SMT-ABC notation to facilitate the model's learning of how each measure is expressed across various tracks.

Furthermore, we observe that the ABC Notation model benefits from additional epochs in the training phase. This suggests that repeated data positively impacts the model's performance. To understand this phenomenon, we introduced the SMS Law for repetitive training with symbolic music data. This law explores how scaling up the training data affects the performance of symbolic music generation models, particularly in terms of validation loss. This investigation aims to provide clear insights into the relationship between data repetition and model efficacy, offering guidance for optimizing model training strategies.

We conducted both objective and subjective evaluations comparing our MuPT model with state-of-the-art models like GPT-4 ChatMusician and MMT, focusing on ABC-notation and MIDI-based approaches. Objectively, MuPT achieved the closest approximation to ground truth, with an average gap of just 0.11, significantly outperforming ChatMusician's 0.48. This seemingly small numerical difference marks a substantial improvement in music generation quality. Notably, MuPT supports multi-track music generation, a feature absent in ChatMusician, enhancing its utility in realistic settings where such complexity is common. In experiments assessing music structure, MuPT surpassed GPT-4 by 17% and ChatMusician by 6% in terms of Intra Similarity and Repetition Rate, demonstrating its superior capability in handling complex musical compositions. Subjective evaluations further validated MuPT's superiority, with over 70% preference ratings against both MMT and GPT-4, underscoring its appeal to human listeners.

Our main contributions are chiefly as follows:

- **Models.** We introduce a series of long-context symbolic music foundation models trained on ABC notation, featuring an extended sequence length of 8,192 tokens, enabling them to accommodate over 90% of the symbolic musical data in our collected dataset. This advancement significantly enhances our ability to process and generate longer, more complex musical compositions.

- **Method.** We propose SMT-ABC notation, a novel approach that seamlessly integrates the characteristics of auto-regressive models with the inherent nature of repeat and music structure with a high compression rate. Our extensive experimentation demonstrates that SMT-ABC not only improves the coherence of generated musical pieces but also preserves the nuanced patterns and structures inherent in music.

- **Scaling Law.** We explore the SMS(Symbolic Music Scaling) Law insights for music modeling based on the ABC notation. We demonstrate that comprehensive music modeling yields superior performance with a positive correlation between model size and metric improvement. We also reveal unique training epoch dynamics in music repetition and performance enhancement.

- **Open Source.** We release a suite of state-of-the-art long-context symbolic music foundation models along with all the intermediate training checkpoints to foster community research and innovation in symbolic music modeling.

## 2 RELATED WORK

**Music Pre-training**    Audio pre-training through the self-supervised learning paradigm has made great progress in speech (Baevski et al., 2020; Hsu et al., 2021; Baevski et al., 2022; Ma et al., 2023b; Yang et al., 2023; Lin et al., 2023), general-purpose audio (Huang et al., 2022; Baade et al.,

2022; Chen et al., 2023; 2024), as well as music (Zhu et al., 2021; Dong et al., 2023; Thickstun et al., 2023; Ma et al., 2023a; Li et al., 2023). Two types of music pre-training have been explored: non-autoregressive discriminative models and autoregressive generative models. Autoregressive generative music pre-training models employ a GPT-style framework to generate music, either in codec (Copet et al., 2024) form or in symbolic form (Thickstun et al., 2023; Dong et al., 2023).

**Data Representation for Symbolic Music** Symbolic music representation formats such as MIDI, Humdrum, and ABC notation offer distinct approaches for representing musical information. Specifically, MIDI, which excels in capturing musical notes and performance, is a popular choice in the music industry and research community(Huang & Yang, 2020; Huang et al., 2019; Lu et al., 2023). However, the complexity and length of MIDI sequences often challenge music models, which limit the preservation of a composition's full continuity. MIDI sequences are typically segmented into shorter fragments, which limit capturing a composition's full continuity. MIDI's encoding of performance nuances can lead to quantization errors and unstable rhythms when being tokenized. Besides, there are some score-level symbolic music, such as MusicXML, MEI, and Lilypond, that are not typically used for model training due to the limitation of large-high-quality data and the length to representing the same music clips Ma et al. (2024) In contrast, ABC notation stands out for its textual simplicity and compactness, making it particularly suited for Natural Language Processing (NLP) techniques. It can be efficiently processed and analyzed using sequence modeling and pattern recognition algorithms similar to those used in language translation and text generation, enabling automated music generation and retrieval(Sturm et al., 2016; Casini et al., 2023; Yuan et al., 2024). However, each soundtrack is recorded sequentially in typical ABC notation, which is different from music performance when each measure within every track is performed simultaneously Yuan et al. (2024). This making multi-track ABC notation somehow incompatible with LLMs on next token prediction. Since if a token is far from the current tokens, it becomes challenging to capture and measure the results effectively.

**Scaling Law** A wide range of research underscores a significant pattern in language model performance, indicating a power-law relationship between model performance and the increases in both the number of parameters and the size of the training data (Kaplan et al., 2020; Hoffmann et al., 2022; Ghorbani et al., 2021). Scaling law plays a pivotal role in advancing large language models (LLMs), offering a framework to predict the optimal configurations for larger models based on the training logs of their smaller counterparts (Gao et al., 2022). The research by Muennighoff et al. (2024), which involves the repetition of the entire pre-training dataset across multiple epochs, presents promising results yet raises questions regarding its effectiveness for musical data. This uncertainty prompts a need for further research into the impact of data repetition strategy by achieving improved outcomes for models engaged in music-related tasks.

| Data Type | Count | Pct. (%) | Avg. Tks |
|---|---|---|---|
| Single Track | 3.5M | 51.2 | 450 |
| 2 Tracks | 605K | 8.7 | 2.0K |
| 3 Tracks | 412K | 5.9 | 3.1K |
| 4 Tracks | 632K | 9.0 | 4.2K |
| 5 Tracks | 362K | 5.2 | 5.2K |
| 6 Tracks | 248K | 3.6 | 6.7K |
| 7 Tracks | 176K | 2.5 | 8.2K |
| 8 Tracks | 149K | 2.1 | 10.1K |
| 9 Tracks | 104K | 1.5 | 10.3K |
| 10 Tracks | 88K | 1.3 | 11.8L |
| 11+ Tracks | 633K | 9.1 | 25.9K |
| Total | 6.9M | 100.00 | 4.53 |

| Genre | Count | Pct. (%) | Type |
|---|---|---|---|
| Pop | 227 | 18.7 | S/M |
| Jazz | 213 | 17.5 | S/M |
| Country | 168 | 13.8 | S/M |
| Rock | 99 | 8.1 | S/M |
| Dance | 75 | 6.2 | S/M |
| Latin | 22 | 1.8 | S/M |
| Folk | 95 | 7.8 | S/M |
| R&B | 111 | 9.1 | S/M |
| Classical | 207 | 17.0 | S/M |
| Total | 1217 | 100.00 | |

Table 1: Training Set Statistics. Pct.(%) refers to the percentage of the specific type of data. Avg. Tks refers to the average number of tokens.

Table 2: Test Set Statistics. Each genre includes both single and multi-track music pieces. Pct.(%) refers to the percentage of the specific type of music. S refers to Single track music, M refers to Multi-Track music.

## 3 DATASET

The dataset used in our empirical study is divided into two parts: a testing set and a training set. The testing set is derived from WIKIMT++(Zhou et al., 2023), which includes 1,010 ABC notation scores from eight music genres (e.g., Pop, Jazz, Rock, R&B, Latin, etc.) along with 12 subjective emotions. Additionally, the test set comprises 207 multi-track classical music pieces manually selected from Bach's compositions. Importantly, none of these pieces overlap with the training set, ensuring that the test set can effectively evaluate the model's performance across diverse musical genres, various emotions and in generating out-of-domain music.

The training set is built from a comprehensive collection, incorporating the Nottingham Music Dataset[1], the ABC tune book of Henrik Norbeck(Ji et al., 2020), the Irishman dataset (Wu et al., 2023), and a private dataset owned by the Central Conservatory of Music (including university library corpus in ABC and other formats that can be converted to ABC like MusicXML, along with internet collections). This rich dataset spans nearly all music genres and includes a diverse range of both single-track and multi-track data. Due to the unavailability of detailed genre metadata for most of the training data, we have not included genre-specific breakdowns in the statistics. We are committed to open-sourcing the training data for research purposes once all datasets are well-organized.

Tables 1 and 2 provide an overview of the statistics for both the testing and training sets, illustrating the number of samples, their distribution, and average token lengths across various categories.

## 4 METHOD

### 4.1 SMT-ABC NOTATION

ABC notation is a widely adopted system for notating music using plain text, and it offers unique advantages when used in conjunction with deep learning models. Its well-structured text format ensures easy preprocessing, efficient data transmission, and scalability of datasets. The diverse collection of tunes and compositions in ABC notation facilitates learning various musical structures and styles. Moreover, ABC notation allows models to generate human-readable outputs, leading to immediate feedback and iterative refinement. These attributes significantly enhance both the efficiency and quality of the training process.

An ABC file is composed of headers following the music notation. The former contains metadata regarding the tune, such as its composer and tempo, while the latter defines the melody. Each note is represented by a letter, with additional symbols conveyingduration, rhythm, and other musical characteristics. Compared to MIDI, another symbolic representation form of music, ABC notation offers a simple and user-friendly text-based format that is easy to read and write, making it accessible for musicians of all skill levels. An example is shown in Figure 1. "V:1" indicates the beginning of the first music track and the lines before it are headers. A tune can consist of one or more tracks, each representing a distinct musical element within the composition. The bars within each track are separated by bar line symbols like vertical lines ("|"), which refer to the standard bar line.

In Yuan et al. (2024), ABC files without any modification are the input of models. However, we found that the models struggle with bar alignment when dealing with multiple tracks. Since a track represents a section or division within a musical composition, such as one of the instrumental or vocal parts in a piece of symbolic music, it is crucial for models to capture the correspondence between tracks. Specifically, this correspondence exists in bars with the same indices, and thus, they should be treated as a series of groups. To this end, we reorganize the tracks as depicted in Figure 2. We concatenate music segments from bars with the same index across all tracks, including their right bar lines. These concatenated elements from different tracks are then enclosed by a pair of a newly introduced symbol "$<|>$", which is not part of the original ABC system. This symbol represents the beginning or the end of a group of bars at the same stage. In cases where a tune contains only one track, each new unit will consist of a single bar. After processing all the bars, we obtain a synchronized version of the music notation, while the headers remain unchanged. The length of the tracks is not always identical due to repetition or other specific musical structures, which are difficult to handle exhaustively. Considering these special samples typically account for

---

[1]https://ifdo.ca/seymour/nottingham/nottingham.html

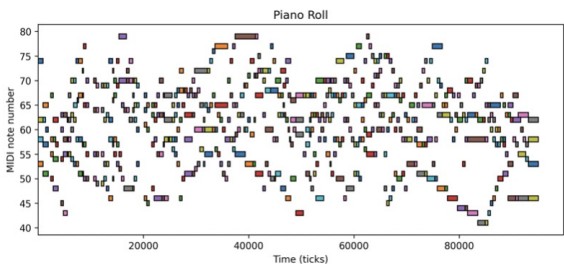

Figure 1: A comparison between MIDI represented by a piano roll (left) and ABC notation (right) of the same music excerpt. MIDI note number represents the pitch and the square's length on the time axis represents the duration of a musical note.

just a small portion (0.01% in our dataset) of the entire dataset, we simply skip them in this scenario. This simple and efficient measure only require 8CPU 1-2 hours to process the whole 33.6B dataset, but provide impressive results in ablation study in subsection B.3.

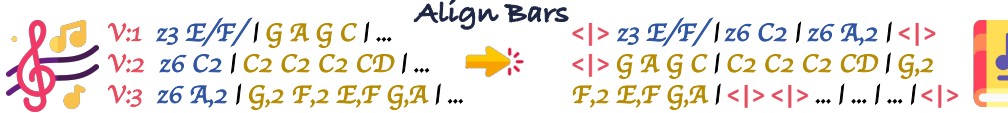

Figure 2: Illustration of synchronized multiple-track ABC notation. Music segments from bars sharing the same index across all tracks, along with their right bar lines, are concatenated to guarantee alignment. The combined elements are then enclosed by a pair of a newly introduced symbol "$<|>$".

## 4.2 TOKENIZER

We chose YouTokenToMe (YTTM) (YouTokenToMe, 2021) framework to develop a tokenizer with a vocabulary of 50,000 tokens, leveraging the Byte-Pair Encoding (BPE) (Shibata et al., 1999) for ABC notation tokenization. This method is instrumental in segmenting the ABC text into manageable units, thereby enhancing the model's ability to interpret and process the input effectively. We do not apply any normalization and dummy prefix to the training corpus, without changing its form or adding extra parts at the beginning. Additionally, a unique symbol "$<n>$"is employed to denote spaces within the ABC text, ensuring accurate space recognition by the model.

| Parameters | 190M | 505M | 1.07B | 1.97B | 4.23B |
|---|---|---|---|---|---|
| Hidden Size | 768 | 1024 | 1280 | 1536 | 2048 |
| # Layers | 12 | 16 | 20 | 24 | 32 |
| # Feedforward dims. | 3072 | 4096 | 5120 | 6144 | 8192 |
| # Heads | 12 | 16 | 20 | 24 | 32 |
| Head Size | 256 | 256 | 256 | 256 | 256 |

Table 3: MuPT model with different model sizes.

## 4.3 MODEL ARCHITECTURE

MuPT utilizes a standard Transformer model architecture (Vaswani et al., 2023) in a decoder-only setup. Models are trained on a context length of 8192 tokens. We list our MuPT model parameter in Table 4.2 and utilize several improvements proposed after the original transformer paper. Below, we list the included improvements:

- **SwiGLU Activation:** The SwiGLU activation mechanism, represented as $(\text{Swish}(xW) \cdot xV)$, is utilized for the MLP (Multi-Layer Perceptron) intermediate activations. This approach signifi-

cantly surpasses traditional activation functions such as ReLU, GeLU, and Swish in performance (Shazeer, 2020).

- **RMSNorm** Each transformer sub-layer, including the attention and feedforward layers, is normalized using RMSNorm as proposed by Zhang & Sennrich (2019)

- **RoPE Embeddings:** In contrast to positional encoding (PE) strategy, we use the Rotary Positional Encoding (RoPE) technique, as developed by Su et al. (2023), aimed at enhancing long-context modeling.

## 4.4 SCALING LAW

The Chinchilla Law, proposed by DeepMind, is a scaling law that provides insights into the training of LLMs. Our experiments reveal that the Chinchilla Law (Hoffmann et al., 2022) provides a good fit for general cases, where moderate models were trained with a moderate amount of data. In this section, we will list several improvements to Chinchilla Law for symbolic music scaling principles on limited training data.

### 4.4.1 OPTIMIZING BASELINE SCALING LAWS UNDER COMPUTATIONAL CONSTRAINTS

A pivotal aspect of scaling laws is the optimization of loss within the bounds of computational feasibility. This is formalized as minimizing the valid loss $L$, subject to constraints imposed by available computational resources ($C$), specifically FLOPs, as denoted below:

$$\arg\min_{N,D} L(N, D) \quad \text{s.t.} \quad \text{FLOPs}(N, D) = C \tag{1}$$

This framework encapsulates the trade-offs between parameters ($N$) and training tokens ($D$), and decision-making processes inherent in scaling models under resource limitations, illuminating pathways to efficiency and efficacy in LLMs training. More details can be found in Appendix A.1.

In this paper, we will use the Chinchilla Law(Hoffmann et al., 2022) and Data-Constrained law(Muennighoff et al., 2024) as baselines. The former is a classical baseline in LLMs' training and the latter is crucial to address the constraints faced in scenarios where the volume of available training data does not meet the ideal requisites. This phenomenon is typical in the music domain. Please refer to A.1.2 for more information.

### 4.4.2 SYMBOLIC MUSIC SCALING (SMS) LAW

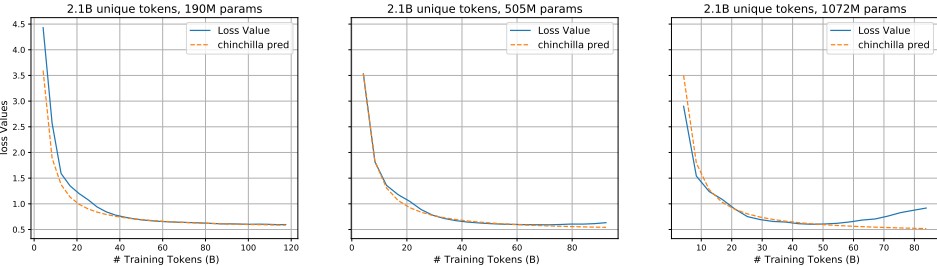

Figure 3: Chinchilla Law prediction and authentic validation loss in the setting with 2.1B unique training tokens for models with 190, 505 and 1072MB.

Figure 3 demonstrates the Chinchilla prediction in yellow lines and the observed loss in blue. We can tell that the Chinchilla law does not provide good results when the data volume $D$ is small when the model just begins the pre-training stage, and when $D$ is large where repeated data provides overfitting. We proposed two terms to address these problems.

**Incorporation of a New Term.** We can observe that when that model parameter is small (e.g. $N = 190M$), the Chinchilla underestimates the loss value and overestimates when the model size is large (e.g. $N = 1072M$). This suggests that the coefficient $B$ in the Chinchilla formula $L =$

$\frac{A}{N^\alpha} + \frac{B}{D^\beta} + E$ shall be relevant to $D$ instead of a constant. To cope with, we incorporate a new term. After that, we proposed another term to predict the early stop points and overfited loss curve:

$$L(N, D) = \frac{d}{N^\alpha \cdot D^\beta} + \frac{A}{N^\alpha} + \frac{B}{D^\beta} + E. \tag{2}$$

Where $\{A, B, d, E, \alpha, \beta\}$ are learned variables fit using the training runs. To address the model's limitations in accurately capturing performance metrics for smaller data sizes, we introduce an additional term, as delineated in Equation 2. This modification aims to refine the model's fidelity, particularly in scenarios characterized by limited data availability. Further details on this modification can be found in the Appendix A.3.1.

**Modelling Overfitting Settings.** Crucially, previous iterations of the model fall short in predicting overfitting, particularly beyond early stopping thresholds. This gap is especially pronounced in the context of Data-Constrained environments, such as music, where open-source data is limited. To this end, we introduce a new component, $L_{overfit}$, to the model, encapsulated in Equation 3, to specifically account for overfitting losses:

$$L(N, D, U_D) = \frac{d}{N^\alpha \cdot D^\beta} + \frac{A}{N^\alpha} + \frac{B}{D^\beta} + E + L_{overfit} \tag{3}$$

where

$$L_{overfit} = GELU\{k_d \cdot D + k_n \cdot \log(N) - k_u \cdot \log(U_D) - k_{in}\} \tag{4}$$

is our overfitting formulation where $\{k_d, k_n, k_u, k_i n\}$ are learned variables for overfitting calibration. For comprehensive insights into the overfitting loss component, please refer to Appendix A.3.2.

**Parameter Fitting and Model Integration.** Initial parameter fitting for $\{\alpha, \beta, A, B, E\}$, and $d$, subsequent linear regression analysis, focusing on the residuals between Equation 2 and empirical observations, facilitates the calibration of overfitting parameters $\{k_d, k_n, k_u, k_{in}\}$ within Equation 4. The integration of these components in Equation 3 not only predicts performance under constrained conditions but accounts for overfitting dynamics, helping to predict the true minimum of loss curve.

## 5 EXPERIMENTS

### 5.1 DATASET & EXPERIMENTAL SETUP

As outlined in section 4.3, we adopt similar model architecture from LLaMA2(Touvron et al., 2023b), including RMSNorm(Zhang & Sennrich, 2019) and SwiGLU(Shazeer, 2020). All the hyperparameters are detailed in Appendix B.1. In the full-scale data setting, we trained models of various sizes (ranging from 190M to 4.23B parameters) on the ABC text corpus, which consists of 33.6 billion tokens derived from a diverse collection of monophonic and polyphonic musical compositions spanning various genres and styles. And the validation set includes 8 pop music genres and classical music, providing good generalization capability for scaling law evaluation. For more information about the corpus, please refer to subsection B.2. In data repetition experiments, we utilized subsets of the corpus, specifically 6.25% and 25% random sampled data. The Adam(Kingma & Ba, 2014) optimizer and cosine learning rate schedules are applied for the training process.

### 5.2 SCALING LAW

#### 5.2.1 EVALUATION METRICS & FITTING METHODOLOGY

We use the $R^2$ value and Huber loss (with the parameter $\delta = 1e - 3$) between the authentic valid loss and predicted valid loss on small models (190M, 505M, 1.07B) to acquire the best scaling law. Then we use the best law to train two large models (with 1.97B and 4.23B). See Appendix A.4 for more details about the two evaluation methods.

We optimized the SMS Law using the L-BFGS algorithm, the same with Chinchilla and Data-Constrained Laws. For more information, please refer to Appendix A.5.

| Paramatic fit | $R^2$ Value (train) ↑ | Huber Loss (train) ↓ | $R^2$ Value (test) ↑ | Huber Loss (test) ↓ |
|---|---|---|---|---|
| Chinchilla law | 0.9347 | 0.0109 | -0.0933 | 0.0080 |
| Data-Constrained law | 0.7179 | 0.0206 | 0.1524 | 0.0071 |
| Equation 11 | 0.9075 | 0.0129 | 0.3114 | 0.0073 |
| Equation 2 | 0.9759 | 0.0102 | 0.8580 | 0.0062 |
| SMS Law | **0.9780** | **0.0085** | **0.9612** | **0.0028** |

Table 4: Comparison of parametric fitting performance of different scaling laws.

### 5.2.2 SMS Law are the Best on the Training Set

The integration of an additional term as delineated in Equation 2, alongside the introduction of a GELU regularization component in Equation 4, collectively underpins the superior performance of the SMS Law, as empirically evidenced by its training set outcomes. This is particularly notable in the context of our parametric fitting performance comparison (see Table 4), where the SMS Law outshines other scaling laws, achieving the highest $R^2$ value (0.9780) and the lowest Huber loss (0.0085) on the training set.

Although Equation 11 does not eclipse the Chinchilla Law in performance metrics, it nonetheless presents a significant improvement over the Data-Constrained Law's $D'$ by leveraging $D''$, which is indicative of a refined approach to managing the constraints posed by data repetition. This nuanced handling of data repetition, inherent to Equation 11, suggests an enhanced generalization capability in such scenarios. Therefore, we culminate it along with other modifications, manifest in the SMS Law in order to enhance model performance and generalization at the same time. In fact, it indeed provides much better results in the test set.

### 5.2.3 Scaled-up Performance using SMS Law

In our SMS Law experimentation under a computational budget of $2 \times 10^{20}$ FLOPs, we initially aim to train a 2.10B (or 1.98B) parameter model across 2.82 epochs on the whole 33.6B dataset per epoch, achieving a loss of 0.5279 (or 0.5280). Engineering constraints necessitated a slight scale-down to a 1.97 billion parameter model, which, intriguingly, showed a minimal loss increase to 0.529 around 2.5 epochs. Contrary to the predictions of SMS Law, the Chinchilla Law suggests optimal performance for a 990M parameter model over 6.1 epochs. Pushing boundaries, we continuously train the 1.07B parameter model and observe overfitting returns beyond 3 epochs, validating the SMS Law's advantages in this context. Further, we train a 4.23B parameter model that underscored the SMS Law's predictive accuracy regarding overfitting risks, affirming its value as a strategic guide in scaling up models effectively within fixed computational constraints, beneficial for efficient model scaling decisions.

In validating SMS Law, we analyze the performance of 1.97B and 4.23B parameter models, detailed on the right-hand side of Table 4. This comparative study highlights the SMS Law's exceptional performance, evidenced by its unparalleled $R^2$ values and minimal Huber Loss on testset as well.

Unlike the Chinchilla and Data-Constrained laws, the SMS Law not only showcase superior predictive accuracy but also demonstrates its efficacy in optimizing neural network scaling within computational constraints. These results affirm the SMS Law's value in guiding scaling strategies for symbolic music, marking a significant advancement in the field. For other advantage of SMT-ABC such as consistency and the subjective evaluation results, please refer to subsection B.3.

### 5.3 Evaluation

#### 5.3.1 Efficiency of Our Training Strategy

To demonstrate the efficiency of our training strategies, we reference the training loss curves in Figure 4. Our comparison spans four different model sizes: 190M, 505M, 1.1B, and 2B. We observed that increasing the training input length from 4096 to 8192 significantly reduces the loss, especially noticeable in the convergence phase. The comparison shows that after aligning data, our training loss slightly decreases compared to the original ABC loss, demonstrating our method's efficiency in improving training for various model sizes. For more discussion on the compression between MIDI, ABC and SMT-ABC, please refer to Appendix C.

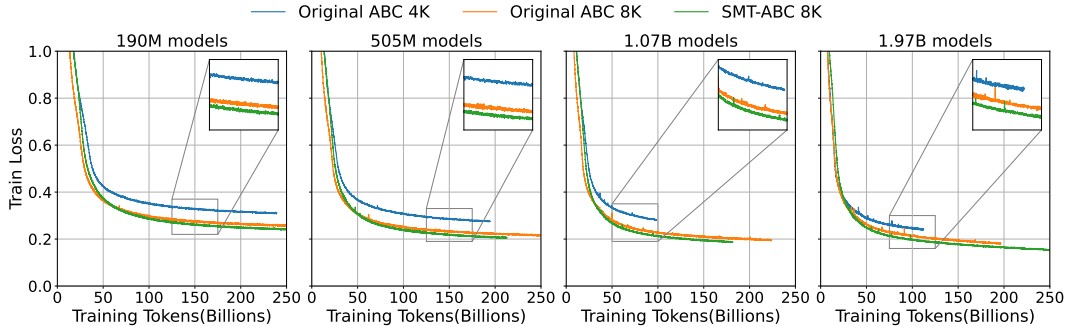

Figure 4: Training Loss for different model sizes and training strategy.

### 5.3.2 OBJECTIVE METRICS OF SYMBOLIC MUSIC ELEMENTS

Following the previous studies on music generation (Dong et al., 2023; Wu & Yang, 2020; Mogren, 2016), we adopt the pitch entropy, scale consistency and groove consistency to evaluate how well the systems can generate music from the perspectives of different musical elements given the first measure. Table 5 shows the mean values of these three metrics, where MuPT achieves overall better performances than other systems compared to the ground truths. For the whole test set, only 51% of samples generated from GPT-4 have the correct ABC notation format. To compare MIDI representation with ABC notations, we incorporate Multitrack Music Transfomers (MMT) (Dong et al., 2023), a MIDI-based music generation model to infer the MIDI data transformed from the ABC notations by abc2midi[2]. Moreover, to compare MuPT with ChatMusician (Yuan et al., 2024), another LLM pre-trained on large-scale single-track (st.) ABC notation data, we separate the single-track samples from our test set and obtain the results in Table 5. MuPT also achieves better results.

### 5.3.3 REPETITION METRICS

**Repetition Rate**   Repetition is significant in evaluating how well-structured the music is. In Table 6, the piece-level average repetition rate of each system is calculated to reveal how often the repeat sign : | appears in a generated set. It appears that 43.7% of the generated samples from MuPT, which is quite close to the ground truth, higher than Chatmusician in single-track data, and much higher than GPT-4. This suggests that MuPT is more likely to generate music with repetition and structure.

| System | PE | SC (%) | GC (%) |
|---|---|---|---|
| GT | 2.708 | 96.80 | 93.46 |
| MuPT-SMT | 2.631 | **97.48** | **93.45** |
| MuPT-Ori. | 2.621 | 98.09 | 93.36 |
| MMT | 2.784 | 95.64 | 91.65 |
| GPT-4 | **2.783** | 97.90 | 95.32 |
| GT(st.) | 2.617 | 98.39 | 93.25 |
| MuPT-SMT(st.) | 2.612 | **98.20** | **93.39** |
| MuPT-Ori.(st.) | **2.619** | 98.16 | 93.49 |
| ChatMusician(st.) | 2.664 | 98.55 | 94.47 |
| MMT(st.) | 2.808 | 95.88 | 91.60 |
| GPT-4(st.) | 2.686 | 99.27 | 95.72 |

Table 5: Mean values of pitch entropy (PE), scale consistency (SC), and groove consistency (GC) for each system. A closer value to the ground truth (GT) is considered better.

| System | ITS | RR (%) |
|---|---|---|
| GT | 0.3729 | 43.5 |
| MuPT-SMT | **0.4193** | **43.7** |
| MMT | 0.1767 | - |
| GPT-4 | 0.3614 | 16.9 |
| GT(st.) | 0.4753 | 59.2 |
| MuPT-SMT(st.) | **0.4507** | **52.6** |
| ChatMusician(st.) | 0.5260 | 40.1 |
| MMT(st.) | 0.2158 | - |
| GPT-4(st.) | 0.4235 | 23.0 |

Table 6: Mean value of intra-texture similarity (ITS) and repetition rate (RR) of each system. The ABC notation string generated by MuPT achieves higher intra-similarity than the GT and GPT-4.

**Intra Similarity**   In addition to the naive repetition rate, we also adopt the methods introduced in Wang et al. (2024) to calculate the intra-similarity of symbolic music in each system. Specifically, a pre-trained VAE from Yang et al. (2019) and Wang et al. (2020) is transferred to compute the

---

[2]https://github.com/xlvector/abcmidi

texture latent for each music piece; the intra-similarity of a music piece is defined as the average value of its texture latent similarity matrix, excluding the diagonal. Since the texture encoder is pre-trained on MIDI data, we transform ABC notations into MIDI format before the latent is obtained. Table 6 shows the mean value of each system's intra-similarity under the first-measure conditioned generation. For the whole test set, MuPT achieves the highest score among all systems, while for the single track, its value is lower than the ChatMusician. Generated pieces of MMT have notably lower intra similarity than MuPT and GPT-4. This result corresponds with the intuition that score-level ABC notation is more capable of generating structured music than performance-level MIDI.

### 5.3.4 SUBJECTIVE EVALUATION

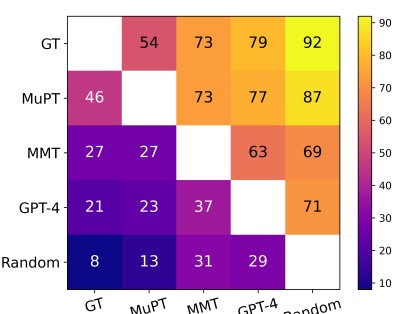

| Model A | Model B | Wins (A/B) | p-value |
|---|---|---|---|
| Human Works | MuPT | 81/69 | 0.4237 |
| | MMT | 109/41 | $4.2249 \times 10^{-6}$ |
| | GPT-4 | 119/31 | $6.6315 \times 10^{-9}$ |
| | Random | 138/12 | $4.4648 \times 10^{-17}$ |
| MuPT | MMT | 110/40 | $4.2249 \times 10^{-6}$ |
| | GPT-4 | 115/35 | $6.6641 \times 10^{-8}$ |
| | Random | 131/19 | $1.3618 \times 10^{-13}$ |
| MMT | GPT-4 | 95/55 | 0.0093 |
| | Random | 103/47 | 0.0001 |
| GPT-4 | Random | 106/44 | $2.6691 \times 10^{-5}$ |

Table 7: Human evaluation of paired completions of musical excerpts generated by different sources given the first bar as the condition. The left is the matrix based on the AB test. Each row indicates the % of times listeners preferred instrumentals from that system compared to those from each system individually (N = 150). Ground truth is denoted by GT. i.e.`77` means that listeners preferred MuPT over GPT-4 in 77% of cases. The right is the absolute win numbers and the corresponding p-value of each pair. P-values are reported by a Wilcoxon signed rank test.

Human assessment should be involved to further testify the objective repetition metrics above. Following Donahue et al. (2023) and and Thickstun et al. (2023), we conduct a subjective listening study to measure the qualitative performance of , we conduct a subjective listening study to measure the qualitative performance of MuPT against the ground truth (GT) and baselines consisting of against the ground truth (GT) and baselines consisting of GPT-4, MMT and random note sequences (Random). Listeners are asked to identify which of two musical excerpts from different sources is more "musical" during the test process. They are also instructed to focus on two aspects of musicality: how consistently the music sounds throughout (e.g., in terms of its melodic contours, rhythmic patterns, and chord progression); and how likely it is that the development of the music follows a clear structure (e.g., verse-chorus division, repetitions). This process is similar to that in and random note sequences (Random). Listeners are asked to identify which of two musical excerpts from different sources is more "musical" during the test process. They are also instructed to focus on two aspects of musicality: how consistently the music sounds throughout (e.g., in terms of its melodic contours, rhythmic patterns, and chord progression); and how likely it is that the development of the music follows a clear structure (e.g., verse-chorus division, repetitions). This process is similar to that in Yuan et al. (2024) and its details are shown in the Appendix and its details are shown in the Appendix E.. Results for all systems are shown in Table 7. Comparing MuPT to GPT-4, listeners prefer music from our system in 79% of cases. A Wilcoxon signed-rank test of these pairwise judgments shows that listeners preferred music from MuPT significantly more than MMT and GPT-4 ($p = 4.2249 \times 10^{-6}$ and $p = 6.6641 \times 10^{-8}$, respectively). We use Fleiss' kappaFleiss & Cohen (1973) to measure the inter-annotator agreement among 15 participants. A kappa value of 0.5807 was obtained, indicating that the participants achieved moderate agreement. This demonstrates the quality of our subjective annotators.

## 6 CONCLUSION

In this paper, we introduce the MuPT series of pre-trained models trained on the largest possible amount of ABC Notation data which set the standard for training open-source symbolic music LLMs. Additionally, we dive deep into the scaling law exploration and propose SMS Law, a specialist in guiding future scaling of symbolic music GPTs. Our results demonstrate that the MuPT series is competitive with mediocre human composers and guarantees state-of-the-art performance on

symbolic music generation. Moreover, MuPT introduces SMT-ABC, reordering the multiple-track original ABC notation format to assist pre-training of MuPT. We conducted comprehensive evaluations of our MuPT model against state-of-the-art models like GPT-4k, ChatMusician and MMT. Objectively, MuPT closely approximated the ground truth, significantly outperforming ChatMusician and demonstrating superior handling of complex musical compositions, including multi-track music, which is absent in ChatMusician. Subjectively, MuPT was preferred by over 70% of human listeners, outperforming both MMT and GPT-4, confirming its effectiveness in realistic music generation scenarios. We believe that the open access of intermediate checkpoints of MuPT, SMS Law, and MuPT series will foster collaboration and innovation within the open-source computational music community, and open the door to the next-generation symbolic music foundation models.

## ETHICS STATEMENT

In designing the MuPT series, we have meticulously adhered to ethical guidelines to ensure fairness, transparency, and the responsible use of AI in music generation. Despite these efforts, ethical challenges such as potential copyright infringement and unintended use of AI-generated music in sensitive contexts remain. We urge the research community to approach these challenges with vigilance and to consider ethical implications carefully when deploying similar technologies. Besides, as suggested by Ma et al. (2024), we strongly urge people to tag music as "AI-generated" for music pieces when uploading to the website, enabling the user to choose (non-)AI-generated music to better protect the rights of musicians. Furthermore, all 15 participants were chosen by the author of this paper, and therefore, there is no additional requirement for ethical approval for the listening test.

## REPRODUCIBILITY

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

# A  SCALING LAW

## A.1  SCALING LAW BASELINE

### A.1.1  ABSTRACTING LOSS METRICS THROUGH THE CHINCHILLA LAW

In this part, we focus on the relationship of loss metrics to various resource budgets in deep learning. It is first put forward by the Chinchilla Law as illustrated in Equation 5. This law posits that both training and evaluation losses can be abstracted as a function of model capacity $N$ and training data size $D$, thus offering an insight to estimate the best combination of resources to be assigned to training.

$$L(N, D) = \frac{A}{N^\alpha} + \frac{B}{D^\beta} + E \tag{5}$$

Here, $L(N, D)$ denotes the loss metric during training or evaluation, which is assumed to exhibit a power-law dependency on $N$ and $D$. The parameters $A$, $B$, $E$, $\alpha$, and $\beta$ are determined by empirical fitting.

### A.1.2  DATA-CONSTRAINED LAW

**Data-Constrained Law: Scaling under Data Limitations.** Complementing the Chinchilla Law, the Data-Constrained Law shows the scaling dynamics of LLMs when facing the data scarcity problem. Here, we strictly refer to the derivation method of Muennighoff et al. (2024). The goal of discovering Data-Constrained Scaling Law is to generalize the expression to multiple epochs where tokens are repeated.

Data-constrained law is defined as:

$$L(N, D, U_D) = \frac{A}{N'^\alpha} + \frac{B}{D'^\beta} + E \tag{6}$$

where

$$\begin{aligned}
N' &= U_N + U_N R_N^\star \left(1 - \exp\left(\frac{-R_N}{R_N^\star}\right)\right) \\
D' &= U_D + U_D R_D^\star \left(1 - \exp\left(\frac{-R_D}{R_D^\star}\right)\right)
\end{aligned} \tag{7}$$

To get a better understanding of the equation, the definitions of each of the above parameters are as follows: Like Chinchilla Law, $N$ is defined as the number of model parameters, and $D$ is defined as the training tokens.

$U_D$ is defined as the number of unique tokens used. For data-constrained law, $U_D$ is computed as $\min\{D, D_C\}$ given a budget of unique data $D_c$.

$U_N$ is defined as the number of "unique" parameters that provide an optimal fit for $U_D$. According to the method mentioned in Muennighoff et al. (2024), given the following learned variables, $\{A, \alpha, B, \beta\, E\}$, the optimal allocation of compute(C) to $N$ and $D$ as follows:

$$\begin{aligned}
N_{\text{opt}}(C) &= G\left(\frac{C}{6}\right)^a \\
D_{\text{opt}}(C) &= G^{-1}\left(\frac{C}{6}\right)^b \\
G &= \left(\frac{\alpha A}{\beta B}\right)^{\frac{1}{\alpha+\beta}} \\
a &= \frac{\beta}{\alpha + \beta} \\
b &= \frac{\alpha}{\alpha + \beta}
\end{aligned} \tag{8}$$

Thus, $U_N$ is equal to $\min\{N_{\text{opt}}, N\}$.

$R_D$ is defined as the number of times the data is repeated. When training for a single epoch, $R_D = 0$.

$R_N$ is the number that the 'unique' parameters are repeated where $R_N = \max\{\left(\frac{N}{U_N}\right) - 1, 0\}$.

$D'$ is defined as the "effective data size": the number of unique data needed to get the same value as repeating $U$ unique tokens for $R_D$ repeats. The derivation process is as followed:

From a conceptual standpoint, the redundancy of data samples diminishes their incremental value in enhancing the model's knowledge base, given the model's prior exposure to said information. This principle underlies the hypothesis that each successive repetition of a sample contributes marginally less to the learning process, as the model has partially assimilated the information contained within the sample through prior iterations. To describe the process of training information loss, we have

$$D' = U + U \sum_{k=1}^{R_D}(1 - \delta)^k = U + (1 - \delta)U\frac{(1-(1-\delta)^{R_D})}{\delta} \tag{9}$$

where $\delta$ is defined as the 'forgetting rate'. Each time a series of tokens is trained on a model, the model learns a $1 - \delta$ fraction information from the optimization process. Assuming that the number of epochs beyond which repeating does not help, the right-hand side goes to to $\frac{(1-\delta)U}{\delta}$, since $\lim_{R_D\to\infty}(1 - (1 - \delta)^{R_D}) = 1$. We define $R_D^\star$ is defined as $\frac{1-\delta}{\delta}$, which is a learned constant. According to Taylor expansion, if $\delta$ is small, we have:

$$e^{\frac{-1}{R_D^\star}} \approx (1 - \delta) \tag{10}$$

Now inserting $\frac{(1-\delta)}{\delta} = R_D^\star$ and $(1 - \delta)^{R_D} = e^{\left(\frac{-1}{R_D^\star}\right)^{R_D}}$ into Equation9, we get our final equation representing the effective data.

As the frequency of encountering repeated tokens diminishes, the benefit gained from processing them also decreases. Hence, the derivation of the $N'$ is similar to $D'$. In this context, there's no need to elaborate further. It should be pointed out that $R_N^\star$ is a learned parameter.

## A.2 ABLITION STUDY ON CONTINUOUS ADAPTATION OF THE DATA-CONSTRAINED LAW.

To enhance the predictive accuracy of the Data-Constrained law (Muennighoff et al., 2024) for continuous domains, we extend the original discrete formulation 11 to accommodate continuous variables, allowing for a more nuanced understanding of data constraints in varied contexts. For an in-depth discussion on the derivation and implications of this continuous formulation, please refer to Appendix A.2.

$$L(N, D, U_D) = \frac{A}{N^\alpha} + \frac{B}{D''^\beta} + E \tag{11}$$

where $k$ is a new parameter to be fit, and $D''$, the adjusted data size, is given by:

$$D'' = \frac{1 - k^{D/U_D}}{1 - k}U_D. \tag{12}$$

The definition of $D'$ in Equation 9 is defined from a discrete version and can not be extended to the case when D is less than $U_D$. So we reform the Equation 9 to

$$
\begin{aligned}
D' &= \frac{1 - (1 - \delta)^{\frac{D}{U_D}}}{\delta} \cdot U_D \\
&= \frac{1 - k_d^{\frac{D}{U_D}}}{1 - k_d} \cdot U_D
\end{aligned}
\tag{13}
$$

where $k_d := 1 - \delta$. This equation is equivalent to equation 10 when $D$ is a positive integer times $U_D$.

We implemented a formula symmetric to $N'$ with $U_N$ and $k_N$. But the calculation results of $k_N \approx 0.999$. To make the formula simple, we use the original $N$ instead of $N'$ in the following formula.

## A.3 MOTIVATION OF SMS LAW

### A.3.1 MOTIVATION OF ADDING POWER OF "$ND$" TERM

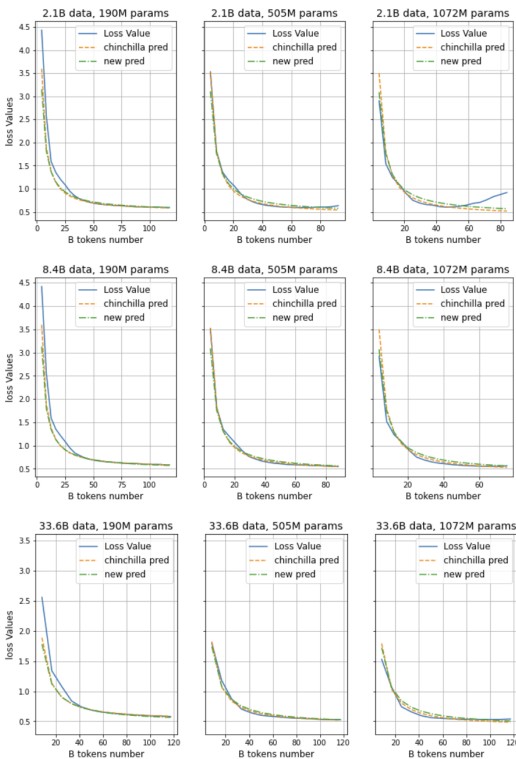

Figure 5: The loss curve, Chinchilla prediction, and Equation11 on 2.1B, 8.4B and 33.6B training data.

In our submission, we present an in-depth analysis of the model's loss dynamics as illustrated in Figure 5, which juxtaposes the empirical loss trajectory (depicted through a blue line) against the theoretical predictions derived from the Chinchilla Law (illustrated by a yellow line) and further contextualized by Equation 11. This comparative study spans three distinct datasets—2.1B, 8.4B, and 33.6B data points—across models of varying capacities: 190M, 505M, and 1.07B parameters, respectively, arranged in a matrix of subfigures with datasets delineated by rows and model capacities by columns.

Observations across all data volumes reveal a nuanced interaction between model and data sizes. Specifically, for smaller datasets and model sizes (190M parameters), predictions consistently underestimate actual loss values, whereas for smaller datasets paired with larger models (1.07B parameters), predictions tend to overestimate. This discrepancy underscores a critical insight: loss reduction accelerates with increasing model size, suggesting a modified loss function, $\frac{A+\epsilon}{N^\alpha}$ over the simpler $\frac{A}{N^\alpha}$

Crucially, the term $\epsilon$ emerges as a function of a single variable $N$, ensuring variability in $\frac{\epsilon}{N^\alpha}$ across each unique model configuration shifting upwards or downwards without changing the shape. The ideal adjustment implies that $\epsilon$ approaches zero for large datasets, yet remains significant for smaller ones, highlighting its dependency on data volume $D$.

In addressing potential overfitting, our strategy focuses on minimizing parameter growth in line with Equation 11. A straightforward approach involves augmenting the loss $L$ into a polynomial encompassing $\frac{A}{N^\alpha}$ and $\frac{B}{D^\beta}$, with Equation 2 introducing an additional term, $\frac{d}{N^\alpha \cdot D^\beta}$, to the existing framework. This refinement, while ostensibly simple, has been shown to yield robust and promising outcomes, exemplifying the efficacy of our proposed modifications in enhancing model performance within the context of scaling laws.

### A.3.2 MOTIVATION OF LINEAR REGRESSION TERM FOR OVERFITTED RESIDUAL

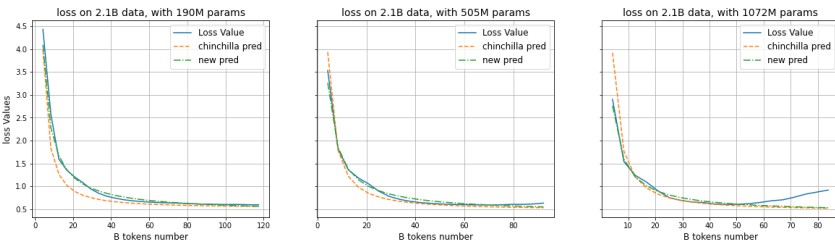

Figure 6: The loss curve, Chinchilla prediction, and Equation 2 (green lines) on 2.1B training data.

Figure 6 offers a detailed exposition on the fidelity of Equation 2 in capturing the loss trajectory across training sets of varied model capacities (190M, 505M, and 1.07B parameters). It is evident from the analysis that the equation adeptly mirrors the empirical loss curve across a broad spectrum of configurations, with the exception of scenarios characterized by concurrently large model sizes and token counts. A notable oversight in the literature is the scant consideration of loss dynamics beyond early stopping points, a consideration of paramount importance in music domain due to the inherent constraints on training data.

In addressing the challenges posed by modelling loss post-early stopping, our investigation delineates two distinct methodologies. The first approach involves the integration of a regularization term within $D''$, aimed at reducing its magnitude beyond the early stopping threshold. Despite its conceptual appeal, this method falls short of providing an adequate fit to the observed data. Alternatively, we explore the augmentation of the loss function $L$ with an additional term, engineered to be negligible when both $D$ and $N$ are minimal, yet incrementally assertive in influencing the loss trajectory after early stopping points. This latter strategy not only aligns more closely with empirical observations but also introduces a nuanced mechanism to accommodate the unique requirements of training in the music domain, thereby extending the utility and applicability of scaling laws within this context.

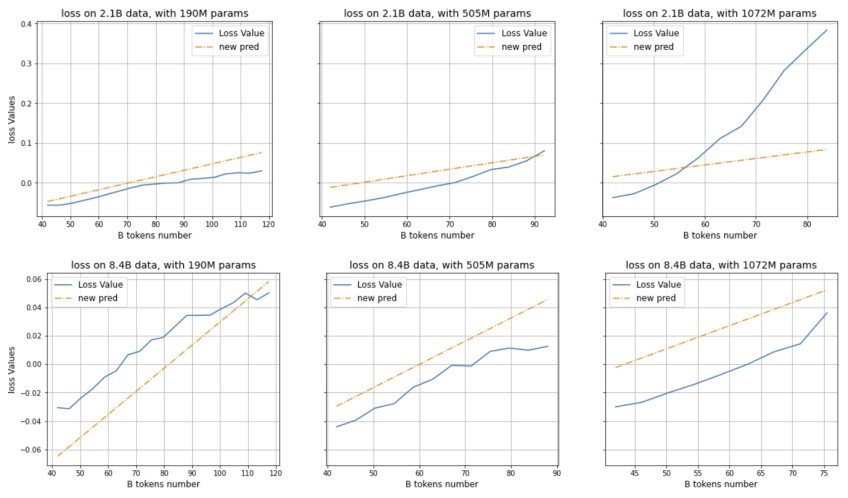

Figure 7: Residule between authentical valid loss and Equation 2 prediction (blue lines), and the linear regression results (yellow lines).

As delineated in Figure 7, the analysis of residuals post the 40 billion token threshold unveils a discernible onset of overfitting, which intriguingly appears to correlate with the model size, data capacity, and the count of unique tokens processed within a single epoch. This overfitting is further characterized by a linear dependency of loss on the total number of processed tokens, coupled with a quasi-linear transition of early stopping points observed across different model capacities (as organized in rows) and magnified across columns.

The progression of model capacities—doubling across rows and quadrupling across columns—illuminates a systematic pattern, suggesting that the early stopping points and consequently, the predicted loss, might be effectively modeled through a linear regression involving dataset size $D$, the logarithm of model capacity $\log(N)$, and and the logarithm of unique tokens per epoch $\log(U_D)$. This observation culminates in the proposition of a regularization term formulated as $k_d \cdot D + k_n \cdot \log(N) - k_u \cdot \log(U_D) - k_{in}$, aimed at encapsulating and mitigating the observed overfitting dynamics.

| Activation Function | $R^2$ (test)↑ | Huber Loss (test)↓ |
|---|---|---|
| ReLU | **0.9786** | 0.0095 |
| LeakyReLU | **0.9786** | 0.0095 |
| GELU | 0.9780 | **0.0085** |
| Tanh | **0.9786** | 0.0094 |
| SELU | 0.9779 | 0.010 |
| Sigmoid | 0.6030 | 0.0700 |

Table 8: Ablition study on the activation function.

In addressing the intricacies of regularization within the context of early model training, especially when considering models of smaller scale (where $U_D$ and $D$ are minimal while $N$ is comparatively large), it becomes imperative to ensure that the regularization term does not adopt a substantially negative value. This stipulation aims to prevent undue penalization at the onset of training, thereby necessitating the incorporation of an activation function that tempers the regularization term's behavior. The Gaussian Error Linear Unit (GELU) function emerges as an apt choice in this scenario. GELU approximates the Rectified Linear Unit (ReLU) function for positive inputs, while also permitting slight negative values with minimal absolute magnitude, thus offering a balanced solution.

Empirical evidence, as detailed in our analysis, underscores the efficacy of applying the GELU function to the regularization term, notably achieving the lowest training loss alongside the second-highest $R^2$ value among the tested models. This finding is particularly salient given the broader magnitude of loss variations relative to $R^2$ values, thereby accentuating the GELU function's suitability for our regularization term. Consequently, the finalized model, incorporating the GELU-modulated regularization term, is depicted through a yellow line in Figure 7. This strategic application of the GELU function not only mitigates the potential for excessive early training penalization but also optimizes the regularization term to enhance model performance effectively.

This approach not only elucidates the linear interdependencies among critical factors influencing model performance but also presents a nuanced regularization strategy designed to enhance model generalizability. Through the integration of this regularization term, we aim to establish a more robust and theoretically informed framework for predicting and managing loss trajectories in large-scale training regimes.

## A.4 EVALUATION METRICS

The R-squared value, also known as the "Coefficient of Determination," is a statistical measure used to evaluate the goodness-of-fit of a regression model. It is defined as:

$$R^2 = 1 - \frac{SS_{\text{res}}}{SS_{\text{tot}}} \tag{14}$$

Where $SS_{res}$ represents the Sum of Squares of Residuals, indicating the total sum of squared differences between the predicted values of the model and the actual observed values, $SS_{tot}$ represents the Total Sum of Squares, indicating the total sum of squared differences between the observed values of the dependent variable and their mean value.

The Huber loss is a type of loss function commonly employed in robust regression models. Unlike the squared error loss, which is sensitive to outliers in the data, the Huber loss is designed to be less affected by outliers. It achieves this by combining the characteristics of both the squared error loss and the absolute error loss. It is defined piecewise by:

$$Huber_\delta(y, f(x)) = \begin{cases} \frac{1}{2}(y - f(x))^2, & \text{if } |y - f(x)| \leq \delta \\ \delta(|y - f(x)| - \frac{1}{2}\delta), & \text{otherwise} \end{cases} \tag{15}$$

For small residuals, it behaves like the squared error loss, whereas for large residuals, it behaves like the absolute error loss. This allows the Huber loss to provide a balance between the two, resulting in a more robust estimation procedure.

## A.5 PARAMETERS FITTING APPROACH

In our study, we adopt a methodology analogous to the Chinchilla Law and the Data-Constrained Law, employing the L-BFGS algorithm—a limited-memory quasi-Newton method—for the optimization of the Huber Loss. This loss function is applied between the logarithm of the predicted loss and the logarithm of the observed (authentic) loss across multiple runs. The objective is to identify the optimal parameters (best_para) that minimize this Huber Loss, formalized as follows:

$$
\begin{aligned}
best\_para &= \min \sum_{runi} Huber_\delta \left\{ \log \left[ \frac{d}{N^\alpha \cdot D''^\beta} + \frac{A}{N^\alpha} + \frac{B}{D''^\beta} + E \right]_i, \log(L_i) \right\} \\
&= \min \sum_{runi} Huber_\delta \left\{ LSE \left[ \log \left( \frac{d}{N^\alpha \cdot D''^\beta} \right), \log \left( \frac{A}{N^\alpha} \right), \log \left( \frac{B}{D''^\beta} \right), \log(E) \right]_i, \log(L_i) \right\} \\
&= \min \sum_{runi} Huber_\delta \left\{ LSE \begin{bmatrix} \log(d) - \alpha \log(N) - \beta \log(D'') \\ \log(A) - \alpha \log(N) \\ \log(B) - \beta \log(D'') \\ \log(E) \end{bmatrix}, \log(L_i) \right\}
\end{aligned}
$$

(16)

where $LSE$ refers to the `log-sum-exp` a numerically stable method to compute the logarithm of a sum of exponentials of inputs. The Huber Loss parameter, $\delta$ is set to $1e-3$, reflecting a stringent criterion for switching between squared loss and absolute loss to ensure robustness in optimization. Additionally, the L-BFGS algorithm's learning rate is configured at $1e-1$, with an update history size of 10 to balance between computational efficiency and the capacity to capture relevant optimization trends.

## A.6 RESULTS OF PROPOSED METHODS WITH EARLY STOPS

| Paramatic fit | $R^2$ Value (train) ↑ | Huber Loss (train) ↓ | $R^2$ Value (test) ↑ | Huber Loss (test) ↓ |
|---|---|---|---|---|
| Chinchilla law | 0.9443 | 0.0073 | -0.0004 | 0.0029 |
| Data-Constrained law | 0.7216 | 0.0189 | 0.1005 | 0.0050 |
| Equation 11 | 0.8356 | 0.0151 | 0.5829 | 0.0045 |
| Equation 2 | 0.9843 | 0.0072 | **0.9866** | **0.00088** |
| SMS Law | **0.9851** | **0.0055** | 0.9864 | 0.00091 |

Table 9: Comparison parametric fitting performance of different Scaling Laws on the curve before early stop points.

From the table, we can see that most of the experimental results increase after we delete the curve after the early stop points. Adding the linear regression still contributes to the performance increase on the training set but provides worse results on test set compared to Equation 2.

# B TRAINING DETAILS & DATASET

## B.1 TRAINING DETAILS

All the models are trained using AdamKingma & Ba (2014), with $\beta_1 = 0.9, \beta_2 = 0.95, eps = 10^{-8}$. We use a cosine learning rate schedule, decay the final learning rate from $3^{-5}$ to $3^{-6}$, with warmup ratio of 0.1. We apply a weight decay of 0.1 and gradient clipping of 1.0. Table 10 shows other training details of each model.

Table 10: Training Details for different ABC format and model settings.

|  | Parameters | Context Length | Trained Tokens | Training Days | Num of GPUs |
|---|---|---|---|---|---|
|  | 190M | 4096 | 119B | 8.4 | 2 |
|  | 505M | 4096 | 97B | 8.4 | 4 |
|  | 1.07B | 4096 | 49B | 8.3 | 4 |
|  | 1.97B | 4096 | 56B | 8.4 | 8 |
| **Original ABC** | 190M | 8192 | 346B | 6.9 | 8 |
|  | 505M | 8192 | 322B | 4.1 | 32 |
|  | 1.07B | 8192 | 223B | 5.4 | 32 |
|  | 1.97B | 8192 | 196B | 8.1 | 32 |
|  | 190M | 8192 | 276B | 5.5 | 8 |
| **SMT-ABC** | 505M | 8192 | 212B | 2.7 | 32 |
|  | 1.07B | 8192 | 181B | 4.4 | 32 |
|  | 1.97B | 8192 | 272B | 11.3 | 32 |
|  | 4.23B | 8192 | 262B | 10.7 | 64 |

## B.2 ADDITIONAL INFORMATION ON TRAINING SET

Metadata for the training dataset The dataset includes a total of more than 1.8 million songs. For the subset of data with genre metadata, the approximate genre distribution is as follows.

| Genre | Number of Songs |
|---|---|
| Pop | 256k |
| Jazz | 107k |
| Country | 49k |
| Rock | 217k |
| Disco | 6k |
| World Music (including Latin) | 47k |
| Folk | 118k |
| R&B, Funk & Soul | 63k |
| Classical | 466k |

Table 11: Genre metadata in Training Set. Each song can have more than one genre tag, and can also have no tag information.

Due to the lack of comprehensive metadata for the entire dataset, we cannot provide precise genre statistics for all songs. Nevertheless, we ensured that the training data is diverse and representative of a wide range of musical styles, enabling robust and comprehensive model training. Besides, to address intellectual property concerns, the datasets are sourced from publicly available repositories, ensuring ethical usage and proper citation. The authors explicitly state that the data is used strictly for research purposes, complying with licenses and copyright laws. The dataset is not redistributed, ensuring adherence to the terms set by the original sources.

## B.3 ABLATION STUDIES ON (SMT-)ABC REPSENTATIONS FOR TRAINING

To validate the effect of the SMT-ABC Notation training strategy, which has previously shown advantages in reduced training loss 5.3.1 and higher consistency rate 5.3.2, we conduct two exper-

iments: the first evaluates measure consistency in multi-track notations, and the second involves subjective evaluations.

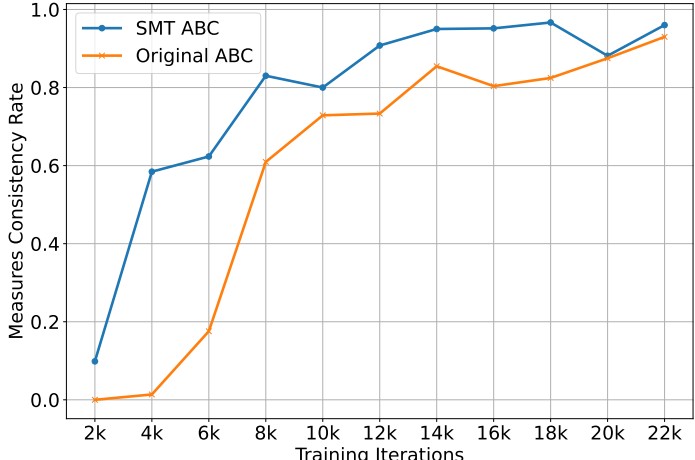

Figure 8: Measure consistency of SMT-ABC and ABC models in different training iterations.

**Measure Consistency**   To assess the measure consistency in generated ABC music sequences, we measure the proportion of sequences where all tracks contain an equal number of measures. Figure 8 illustrates that the sequences generated by the SMT-ABC model demonstrate a significantly higher consistency rate compared to those generated by the model trained on Original-ABC notation. This suggests that the SMT-ABC notation facilitates models to maintain structural uniformity across different tracks, which is critical for ensuring the coherence and usability of the generated compositions in practical applications.

**Objective and Subjective Evaluation**   In Table 5, MuPT-SMT and MuPT-Ori. represent the SMT-ABC notation and Original-ABC notation respectively. The results show that mostly SMT-ABC performs better than Original-ABC. Meanwhile, we also conduct the AB test of all multi-track samples in the test set between these two systems and it shows listeners prefer music from SMT-ABC in 53% of cases than Original-ABC. ($p = 2.7265 \times 10^{-6}$).

## C   FURTHER DISCUSSION ON SYMBOLIC MUSIC TOKENIZATION

While there are many representations for symbolic music besides ABC notation and MIDI, including some adaptations of MIDI such as REMI, etc. (Ma et al., 2024), the following discussion focuses on the vanilla setting of MIDI, ABC notation and our proposed method SMIT-ABC. We leave the ablation study on different symbolic music tokenizers to the future work

### C.1   THE ALIGNMENT AND DIFFERENCE BETWEEN ABC NOTATION AND MIDI

ABC is the music staff/stave in terms of natural language notations, including tune info (e.g. title, composer, meter, key, pitches, rhythms, bar lines & repeats, instrument, etc. The "K: Bb" you mentioned in ABC means the key of the music piece is Bb, which is the same as the key of the MIDI note in the piano roll and is related to the overall pitch of the MIDI note in the piano roll. And " | F2 z G | " corresponds to one music bar that includes a quarter note at pitch F, an eighth-note/quaver rest, and an eighth-note at pitch G. MIDI start and stop times are absolute time slots in seconds and do not include quantized beat/bar information. This given measure corresponds to two MIDI note rectangles on the chart that correspond to the pitch and time (the rest is not shown).

The velocity, control information, and also sound effects that are included in MIDI are not included in ABC notations. ABC notation includes bar and note quantization information (how many beats v.s. how many seconds per note), key, and music/tune structure such as repeat sign, etc., that MIDI does not include. In general, MIDI can keep more information on the styles of performers with acoustic information, and ABC notations are better with modelling repeats and structures, leading to better compression rate and more suitable for LLM training.

## C.2 Advantage of ABC notation

ABC notation offers significant advantages over MIDI in terms of compression ratio, which directly impacts model efficiency and reduces training costs. As highlighted in ChatMusician, ABC notation achieves an average of 288.21 tokens per song and 5.16 tokens per second, requiring roughly 38% fewer tokens than various MIDI-based representations. This remarkable reduction in token count stems from ABC's ability to efficiently encode symbolic musical repetition using concise notations such as repeat signs (| : and : |). These symbols effectively represent patterns spanning several seconds to minutes. By reducing sequence length, ABC not only lowers computational overhead but also simplifies the learning complexity, making it an ideal format for music-related tasks.

It is undeniable that MIDI offers more detailed dynamic information compared to leadsheet formats like ABC. However, a major drawback of MIDI is that, within a limited context length, it may not be possible to encode an entire song. While hierarchical modeling methods, such as Whole-Song Hierarchical Generation of Symbolic Music Using Cascaded Diffusion Models, have been proposed to address this issue, scaling up such approaches can be challenging. In contrast, ABC notation is compatible with natural language symbols and naturally encoded structures like repetition, benefitting training efficiency, convergence speed, and modelling the overall structure of a song with NLP codebase. Moreover, ABC notation integrates seamlessly with text tokenizers, making it an efficient choice. After carefully weighing the pros and cons of both formats and considering the characteristics of our collected dataset, we decided to use ABC notation.

## C.3 Comparsion between SMT-ABC and ABC

Our proposed SMT-ABC notation effectively addresses the alignment challenges in multitrack ABC notation, significantly improving training loss and overall model performance.

One major issue is the measure consistency problem. As shown in Appendix Figure 8, this figure illustrates the measure consistency scores for SMT-ABC and standard ABC notation under the same training iterations and model structures. The results demonstrate that as training progresses, SMT-ABC quickly achieves consistent measures across tracks, indicating effective alignment and coherence. In contrast, models trained solely with ABC notation struggle to maintain measure consistency across tracks, even as training iterations increase. This highlights the advantage of SMT-ABC in addressing alignment issues and improving overall musical structure.

The concepts of bar coherence and track coherence are useful but challenging to evaluate directly. Instead, we propose the concept of the closeness of related tokens. Since this is a causal model for pretraining, the later tokens in the sequence are less influenced by earlier tokens as the tokens gap increases. The purpose of designing the SMT-ABC notation is to ensure that bars played in each track at the same time are generated as closely as possible. Thus, we focus on discussing the token gaps between different tracks.

In our training dataset, which aligns with the natural distribution of ABC notation, the average tokens per bar is 3.38 tokens, while the average bars per track is 77.40 bars. This means that the benefits of reducing token distances between tracks significantly outweigh those of reducing token distances within a track.

For example, we define the most related bars as: the bar with the same index in a different track, and the next bar in the same track. In a 4-track piece of music: the token distance between Track 1, Bar 1 and Track 4, Bar 1 in standard ABC notation is approximately $77.4 \times (4 - 1) = 232.2$ tokens. And the token distance to the next bar within the same track is much smaller ( 3.38 tokens ).

In contrast, with SMT-ABC notation: the token distance between Track 1, Bar 1 and Track 4, Bar 1 is reduced to: $3.38 \times (4 - 1) = 10.14$ tokens. However, the token distance to the next bar in the same track increases slightly to $3.38 \times 4 = 13.52$ tokens . This design in SMT-ABC notation brings related bars closer together while slightly increasing the distance for less related bars. This shift in token distribution improves the closeness of related tokens, enhancing the overall coherence and ultimately boosting the model's final performance.

## C.4 APPLICATIONS OF ABC NOTATION

Though MIDI is widely used in the music industry, ABC has a strong potential for application in music generation. Leadsheet-based multitrack symbolic music generation is a promising yet underexplored area. Leadsheets effectively compress the semantic information of music, facilitating whole-song-level modeling. Besides, ABC notation is designed with natural language symbols that can be easily combined with natural language. This approach could guide the text-to-music generation models similar to Suno's framework integrating multimodal capability easily. Furthermore, it is also worth noting that audio can be generated from ABC leadsheets by using a rendering model in various style. For example, Seed-Music: A Unified Framework for High Quality and Controlled Music Generation demonstrates this approach, enabling the generation of high-quality audio based on ABC notation. This allows for the inclusion of performance characteristics and timbre while maintaining the capability to model entire songs with ease.

## D SIMILARITY BETWEEN TRAININGSET AND INFERENCE SAMPLES

We conducted experiments to investigate whether the music generated by the model exhibits any copying from the training data. Specifically, we sampled 750 pieces of music with 1-15 tracks and 100 pieces with more than 15 tracks from the training set. For these samples, we used their headers and the first bar as prompts and calculated the similarity between the model-generated music and the original music from the training set. The similarity was measured using three metrics: n-grams (character-based segmentation), n-gram-music (note/chord-based segmentation), and the longest common subsequence(LCS) between the two music sequences. The results are shown in the table below.

| N-gram(n=4) | N-gram-Music(n=4) | LCS |
| --- | --- | --- |
| 0.19 | 0.08 | 0.29 |

Table 12: Similarity between Training set samples and model generation.

The results suggest that the model is capable of generating novel sequences rather than simply copying training data. The low n-gram and n-gram-music scores, paired with the moderate LCS score, indicate that the model retains stylistic or structural influences from the training set without overfitting to exact examples.

## E HUMAN ASSESSMENT

### E.1 ADDITIONAL INFORMATION OF HUMAN ASSESSMENT

We use webMUSHRA toolkit (Schoeffler et al., 2018) to conduct a web-based subjective listening AB-test. About the music background of participants, 30% of them are beginners, 40% are intermediates, 25% are advanced and 5% are experts. During the listening test, we ask the participants to choose the better one between a pair of music excerpts generated from two randomly selected different systems from *GT, MuPT, GPT-4, MMT and Random* by considering the "Musicality" which indicates the overall perceptive quality of the music. Participants are encouraged to make a choice by refering to the guidelines below:

- How consistent the music sounds as a whole (e.g., in terms of its melodic contours, rhythmic patterns, and chord progression).
- How likely the development of the music follows a clear structure (e.g. verse-chorus division, repetitions).
- If you cannot understand the two guidelines above, just choose the one from A and B you prefer.

We use Fleiss' kappa (McHugh, 2012) value to measure the inter-annotator agreement among 15 participants. A kappa value of 0.5807 was obtained, indicating that the participants at least achieved moderate agreement (if the value is 1, the perfect agreement is obtained) by referencing Table 13 with only two participants. This demonstrates an acceptable quality of our subjective annotators.

Table 13: Fleiss' kappa metrics interpretation

| Subjective example: only for two annotators, on two classes (Landis JRKoch, 1977). |
| --- |
| 0  Poor agreement |
| 0.01 – 0.20  Slight agreement |
| 0.21 – 0.40  Fair agreement |
| 0.41 – 0.60  Moderate agreement |
| 0.61 – 0.80  Substantial agreement |
| 0.81 – 1.00  Almost perfect agreement |

### E.2    CASE STUDY ON GENERATION RESULT OF MUPT AND GPT-4

We show two pairs of examples generated by MuPT and GPT-4. Each pair of examples are generated given the same first bar. For all outputs in ABC notation, we export the corresponding human-readable music sheets here. The single-track samples of MuPT and GPT-4 are shown in Figure 9 and Figure 10 and the multi-track samples are shown in Figure 11 and Figure 12. Compared with GPT-4, the MuPT generates a better-structured single-track sample with more harmonious chords and repetition symbols in separated theme and a multi-track sample with a closer relationship and more similar progression.

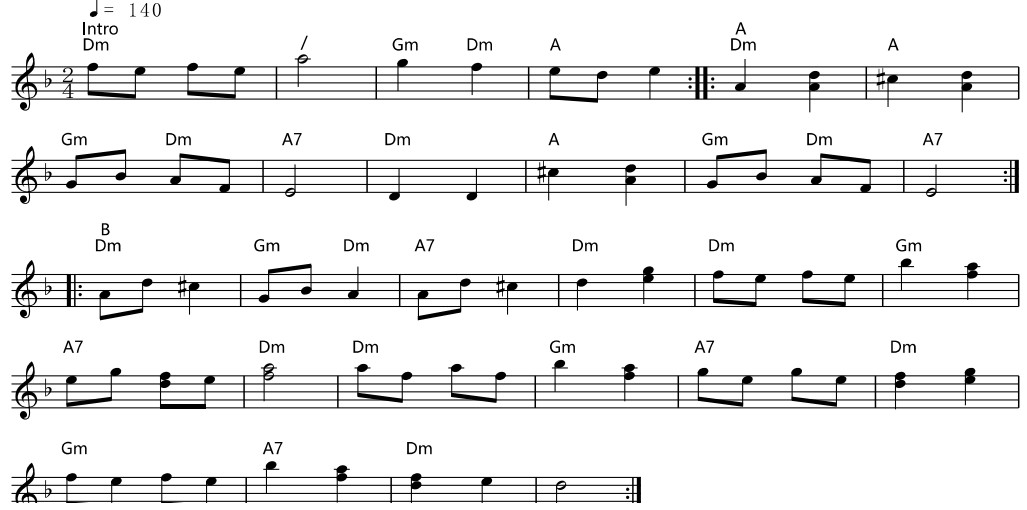

Figure 9: MuPT single-track example.

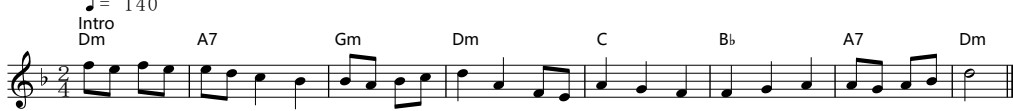

Figure 10: GPT-4 single-track example.

## F    LIMITATIONS

In this paper, we introduce the MuPT series, comprising pre-trained models dedicated to symbolic music generation. These models set a new standard for training open-source symbolic music GPT. However, our models primarily accept input in ABC notations and lack the capability for interactive generation based on human instructions, unlike systems such as Chat Musician Yuan et al. (2024)..

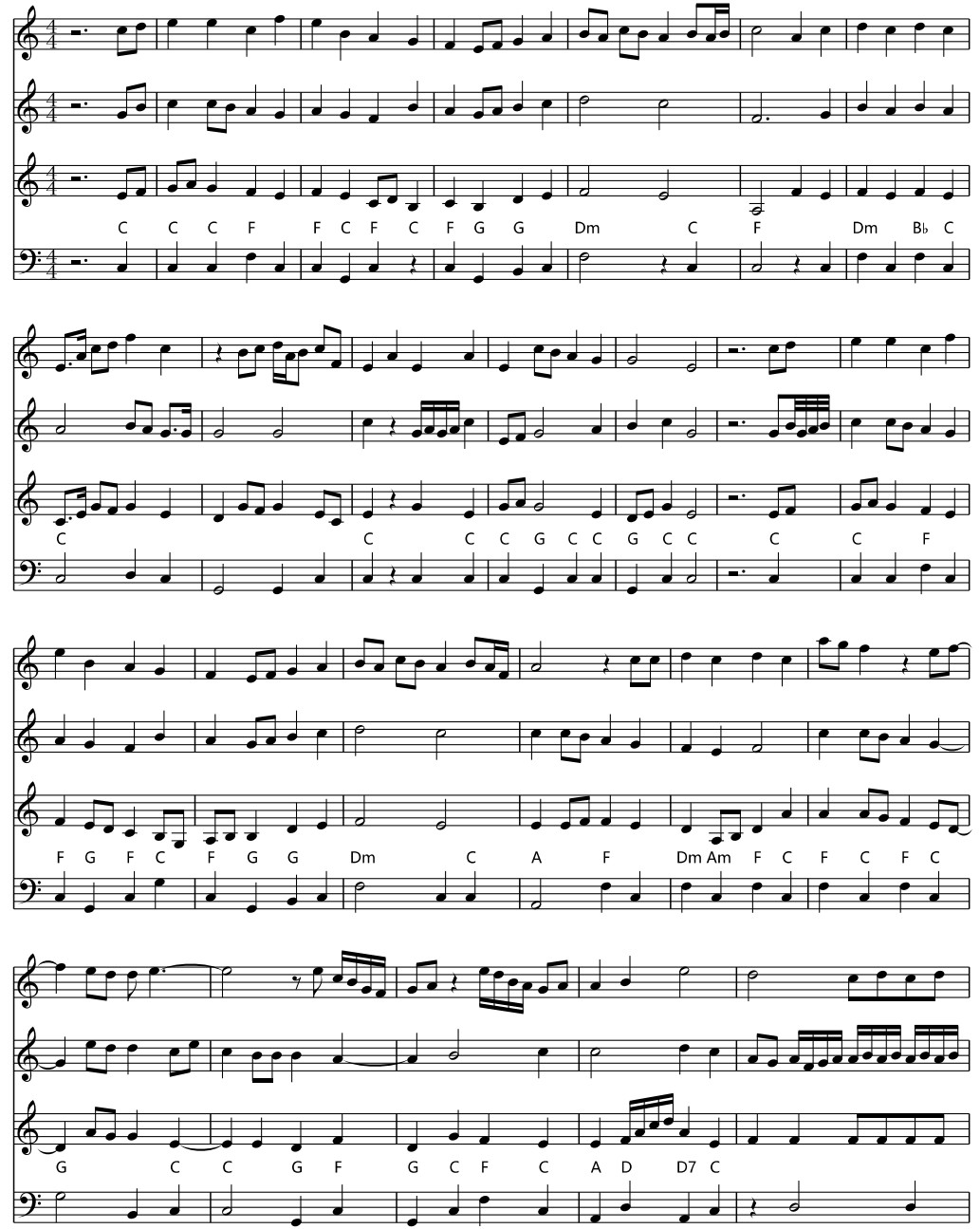

Figure 11: MuPT multi-track example.

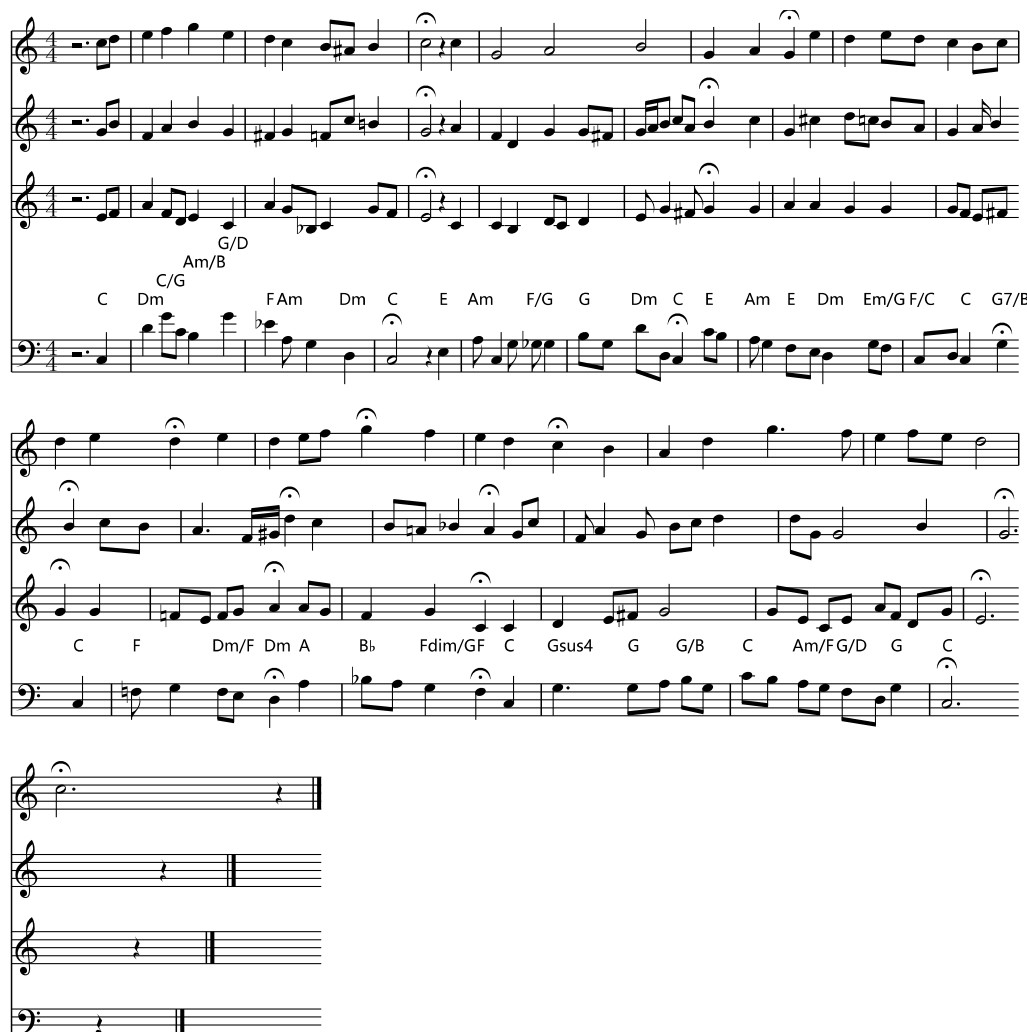

Figure 12: GPT-4 multi-track example.

