# OpenReview forum: "MuPT: A Generative Symbolic Music Pretrained Transformer"
_ICLR.cc/2025/Conference — ICLR 2025 Poster_

### Official Review · Reviewer_PkBU · 2024-10-21

**Soundness:** 3
**Presentation:** 3
**Contribution:** 2
**Rating:** 6
**Confidence:** 4

**Summary:**

The paper introduces a Synchronized Multi-Track ABC to improve coherence across tracks in music generation, addressing the limitations of traditional MIDI formats. Combining with the LLM,  MuPT outperforms other models in terms of repetition rate and structure in multi-track music generation. The paper also investigates the Symbolic Music Scaling Law, showing improvements in performance as model size and training data scale up. However, the novelty of the model is limited, and the effect of the proposed SMT-ABC representation seems not very significant.

**Strengths:**

1. This paper introduces SMT-ABC, a novel representation that improves coherence across multi-track music generation.
2. The investigation into SMS Law offers valuable insights into the relationship between data repetition and model performance, providing practical guidelines for optimizing large-scale music generation models.

**Weaknesses:**

1. There are some mistakes like missing reference targets (e.g. in Section 5.1 ‘please refer to ??’).
2. Although the SMT-ABC representation is interesting by adding a special token ‘<|>’ and changing the sequence of the original ABC notations, the novelty of this paper seems limited, for the model architecture is mostly based on existing works.
3. Changing the sequence of the original ABC notation from bar-sequence to track-sequence may preserve coherence across multiple musical tracks. However, what about the coherence in a track? The contents inside a bar seem to be adjusted to the very back of the input sequence because SMT ABC aligns the tracks first. Besides, from Table 5, the improvement of SMT-ABC seems very slight compared with the original ABC.

**Questions:**

1. What is the model scale of SMT-ABC and Original-ABC in Figure 8 in Appendix B.2?
2. Since GPT-4 is not a model for symbolic music generation, what is the motivation to compare with GPT-4?

---

> ### Author Response · Authors · 2024-11-24
>
> ## Cons
>
> #### **Mistake**
> Thank you for pointing out the typos in our manuscript. If the paper is accepted, we will ensure that these issues are corrected in the camera-ready version.
> #### **Novelty of this paper**
> - **Leadsheet-based multitrack symbolic music generation** is a promising yet underexplored area. Leadsheets effectively compress the semantic information of music, facilitating whole-song-level modeling. In the future, this approach could guide audio music generation models similar to Suno's framework and seamlessly integrate text-based leadsheet generation into the framework of large language models (LLMs).
>
> - Our proposed **SMT-ABC notation** effectively addresses the alignment challenges in multitrack ABC notation, significantly improving training loss and overall model performance.
>
> - To the best of our knowledge, we have collected the **largest ABC notation dataset** and successfully scaled model training on this dataset. For the first time, we have derived a scaling law for symbolic music, laying a solid foundation for integrating musical creativity into general artificial intelligence systems in the future.
>
> - While scaling laws have been extensively studied in other domains, limited research exists on scaling laws specific to ABC notation. Due to the relatively small amount of available ABC-notation data, repetitive training is necessary to achieve meaningful performance. Importantly, ABC notation differs significantly from standard language data, both in structure and representation. Our work addresses this gap by providing insights into scaling laws and repetitive training for ABC data, offering a significant contribution to this underexplored area.
>
> #### **Changing sequence of ABC**
>
> One major issue is the **measure consistency problem**. As shown in Appendix B.2, Figure 8, this figure illustrates the measure consistency scores for SMT-ABC and standard ABC notation under the same training iterations and model structures. The results demonstrate that as training progresses, **SMT-ABC quickly achieves consistent measures across tracks**, indicating effective alignment and coherence. In contrast, models trained solely with ABC notation struggle to maintain measure consistency across tracks, even as training iterations increase. This highlights the advantage of SMT-ABC in addressing alignment issues and improving overall musical structure.
>
>
> #### **Track coherence vs. Bar coherence**
> There are two ways to discuss this issue.
> 1. the final performance of SMT-performance. As shown in Table 5 and the final loss As shown in Figure 4. the training loss and the evaluation metrics are well shown that the performance for SMT-ABC is better than ABC-notation.
>
> 2. The concepts of **bar coherence** and **track coherence** are useful but challenging to evaluate directly. Instead, we propose the concept of the **closeness of related tokens**. Since this is a causal model for pretraining, the later tokens in the sequence are less influenced by earlier tokens as the tokens gap increases. The purpose of designing the SMT-ABC notation is to ensure that bars played in each track at the same time are generated as closely as possible. Thus, we focus on discussing the token gaps between different tracks.
>
> In our training dataset, which aligns with the natural distribution of ABC notation, the **average tokens per bar** is **3.38 tokens**, while the **average bars per track** is **77.40 bars**. This means that the benefits of reducing token distances between tracks significantly outweigh those of reducing token distances within a track.
>
> For example, we define the **most related bars** as:
>    - The bar with the **same index in a different track**.
>    - The **next bar in the same track**.
>
>
> In a 4-track piece of music:
> - The token distance between **Track 1, Bar 1** and **Track 4, Bar 1** in standard ABC notation is approximately:
>   \( 77.4 $\times$ (4-1) = 232.2 tokens \).
> - The token distance to the **next bar** within the same track is much smaller:
>   \( 3.38 tokens \).
>
> In contrast, with SMT-ABC notation:
> - The token distance between **Track 1, Bar 1** and **Track 4, Bar 1** is reduced to:
>   \( 3.38 $\times$ (4-1) = 10.14 tokens \).
> - However, the token distance to the **next bar** in the same track increases slightly to:
>   \( 3.38 $\times$ 4 = 13.52 tokens \).
>
> This design in SMT-ABC notation brings **related bars closer together** while slightly increasing the distance for less related bars. This shift in token distribution improves the closeness of related tokens, enhancing the overall coherence and ultimately boosting the model’s final performance.

---

> > ### Author Response · Authors · 2024-11-24
> >
> > ## **Questions**
> >
> > ```
> > What is the model scale of SMT-ABC and Original-ABC in Figure 8 in Appendix B.2?
> > ```
> >
> > The model size in Figure 8 is 1.97B
> >
> > ```
> > Since GPT-4 is not a model for symbolic music generation, what is the motivation to compare with GPT-4?
> > ```
> >
> > Because GPT-4's training data also includes ABC-notation, it can generate entire ABC scores when prompted with the header of an ABC-notation score. However, our work differs in a critical way: our tokenizer is specifically designed for ABC-notation, and our model is trained exclusively on ABC-notation-based data.
> >
> > Despite being a smaller model with only 2 billion parameters, our specialized design allows us to generate higher-quality scores compared to GPT-4, which is a general-purpose model. This highlights the advantages of a domain-specific approach tailored for music generation.

---

> > ### Comment · Reviewer_PkBU · 2024-11-25
> > **Feedback to the authors**
> >
> > Thanks for the authors’ detailed response. SMT-ABC representation makes sense through your explanation.

---

### Official Review · Reviewer_f4e7 · 2024-10-31

**Soundness:** 3
**Presentation:** 3
**Contribution:** 2
**Rating:** 8
**Confidence:** 3

**Summary:**

The paper "MuPT: A Generative Symbolic Music Pre-Trained Transformer" explores the use of Large Language Models (LLMs) for symbolic music generation, with a focus on leveraging ABC notation over the traditional MIDI format. The authors propose the Synchronized Multi-Track ABC Notation (SMT-ABC Notation) to improve coherence across musical tracks. They present models capable of handling up to 8192 tokens and introduce the Symbolic Music Scaling Law (SMS Law) to optimize the training of these models. Their MuPT model outperforms GPT-4 and ChatMusician in both objective metrics and subjective evaluations of music generation, and they have open-sourced their models to encourage community-led research.

**Strengths:**

The introduction of SMT-ABC Notation represents a novel approach to enhancing the coherence of multi-track symbolic music generation. The use of ABC notation as a foundation for LLMs in music generation, instead of the more commonly used MIDI, adds an original perspective to the field. The concept of the Symbolic Music Scaling Law (SMS Law) is also a significant contribution, offering new insights into the training dynamics of symbolic music models.
The paper is well-structured, with clear explanations of the SMT-ABC Notation and the SMS Law.
The open-sourcing of the models and intermediate checkpoints is a valuable contribution, enabling further exploration and innovation in symbolic music modeling by the broader community.

**Weaknesses:**

1. The experimental design is questionable. SMT-ABC is the only methodological innovation in this paper, yet there is insufficient comparative experimentation to explore how SMT-ABC actually improves performance. Models like GPT-4 are not specifically designed for symbolic music generation, and it's predictable that the proposed model would perform better without even conducting experiments. Were the other models in the comparative experiments, such as GPT-4 and MMT, trained or fine-tuned on the SMT-ABC dataset? If not, it may not demonstrate the true effectiveness of the algorithmic innovation of SMT-ABC.

2. The Scaling Law experiments are detailed, but they primarily validate some emergent performance issues that have been verified in other domains. It does not clearly indicate whether the issues in the music domain can be resolved through the Scaling Law, which suggests that the detailed Scaling Law experiments may not be significantly relevant to the problems this paper aims to address.

3. (1) The training data lacks specific metadata, such as detailed categorization and style of music, which may limit the model's generalization ability in specific styles or types of music. (2) The paper requires adjustments for the alignment of different tracks, and the model may still have certain limitations in generating more complex musical structures, such as polyphonic music. (3) There is no in-depth analysis of the detailed differences between different model architectures, their respective strengths and weaknesses in handling symbolic music, which may make it difficult for readers to understand the specific improvements of MuPT over other models. (4) Subjective factors such as the musical background and preferences of listeners may affect the results of subjective evaluations, and more diverse assessments might provide more comprehensive conclusions. (5) Although experiments show that repeated data can improve model performance, it may lead to overfitting in some cases. Although the paper discusses early stopping and overfitting, more experimental support is needed for the effectiveness of these explanations.

**Questions:**

1. The paper's experimental design is flawed due to insufficient comparison with other models to validate SMT-ABC's performance improvement. It's unclear if models like GPT-4 and MMT were trained on the SMT-ABC dataset, questioning the true effectiveness of SMT-ABC's innovation.

2. The Scaling Law experiments, while detailed, do not clearly show their relevance to music domain issues, suggesting they may not be significantly pertinent to the paper's objectives.

3. (1) The training data lacks metadata, potentially limiting model generalization. (2) The model may struggle with complex musical structures like polyphony. (3) There's a lack of analysis on model architectures' strengths and weaknesses in symbolic music. (4) Subjective evaluations might be biased, necessitating more diverse assessments. (5) Repeated data may improve performance but could also cause overfitting; more experimental evidence is needed to support claims on overfitting prevention.

---

> ### Author Response · Authors · 2024-11-24
>
> Thanks for your inputs and feedback. This is super valuable for us.
> Questions about SMT-ABC
>
> The name __SMT-ABC__ indicates that this model is specifically designed for ABC-notation. While it could potentially be adapted for MIDI data, this does not align with the purpose of our paper. Our focus is on ABC-notation and how to fully leverage it within large models. SMT-ABC has already demonstrated its ability to outperform models trained on the original ABC data, showcasing its effectiveness.
> We believe SMT-ABC could also be beneficial for enhancing models like GPT-4 and MMT. However, MMT does not release their training pipeline, and GPT-4 is not suitable for direct training or fine-tuning, as it is a closed-form model. Due to these limitations, we are unable to conduct further experiments in this regard.
>
> __Scaling law__
>
> We appreciate the reviewer’s feedback and clarify that SMS Law is essential for symbolic music LLMs:
>
> Data-Constrained Setting for Music Domains: Music modeling often faces dataset sparsity, where existing laws like the Data-Constrained Law (NeurIPS2023 best paper) fail on prediction. Symbolic Music Scaling (SMS) Law incorporates a novel "overfitting" term tailored for symbolic music and other data-constrained domains, achieving superior predictive accuracy in such scenarios.
>
> Scaling Insights: SMS Law predicts that models beyond 2B parameters may not yield better results due to dataset constraints. Our experiments confirm this, with the 2B model outperforming the 4B model, optimizing efficiency and quality for developping symbolic music LLMs.
>
> ABC Notation Specialization: SMS Law uniquely supports symbolic music tokens like ABC Notation, with learned parameters and mathematical format significantly different from related domains like speech, audio or natural language symbols, making it a critical advancement for music-specific scaling.
>
> These contributions highlight SMS Law’s methodological and practical relevance to symbolic music LLMs.
>
> __Reply to your additional questions__
> - metadata for the training dataset
> The dataset includes a total of more than 1.8 million songs. For the subset of data with genre metadata, the approximate genre distribution is as follows:
>
> | Genre	| Number of Songs |
> | ---- | -------- |
> | Pop	| >256k |
> | Jazz	| >107k|
> | Country	| >49k|
> | Rock	| >217k|
> | Disco	| >6k|
> | World Music (including Latin)	| >47k|
> | Folk	| >118k|
> | R&B, Funk & Soul	| >63k|
> | Classical	| >466k|
>
> Due to the lack of comprehensive metadata for the entire dataset, we cannot provide precise genre statistics for all songs. Nevertheless, we ensured that the training data is diverse and representative of a wide range of musical styles, enabling robust and comprehensive model training.
> Besides, to address intellectual property concerns, the datasets are sourced from publicly available repositories, ensuring ethical usage and proper citation. The authors explicitly state that the data is used strictly for research purposes, complying with licenses and copyright laws. The dataset is not redistributed, ensuring adherence to the terms set by the original sources.
> - polyphonic music generation
> Our model can generate polyphonic music. Plz refer to our submission materials and Figure 11, which is the multi-track examples generated by MuPT.
> - Architecture comparison
> As our methods is a pretraining model, I would like to say this is resource extensive, and but we do make a research before training our models. ABC-notation can be generated by GPT-4, Chatmusician is also discussed is benefits with ABC-notation and why this suitable for the Large lanuage model. We should use the widely used model architecture, which is LLama-based architecture, to prove that the LLM can be applied to ABC-notation, And this kind of model is suitable for ABC-notation generation tasks.
> - Subject comparision
> Details are shown in Appendix C and Ethics Statements parts.
> Here are some details about the participants background, we devide in four categories, beginner, itermediate, advanced and expert. For each category of participants and their propertion are shown as follows:
> Beginner: you have little to no experience in music and is just starting to learn or explore it. 30.0%
> Intermediate: you have some knowledge and skills in music, such as being able to play an instrument, read sheet music, or understand basic music theory. 33.3%
> Advanced: you have a high level of proficiency in music. You have extensive experience in playing an instrument, performing in ensembles, composing music, or have in-depth knowledge of music theory. 26.7%
> Expert: you are highly skilled and knowledgeable in music, possibly with professional training or experience. You may be a professional musician, music teacher, or have a deep understanding of advanced music theory and performance techniques. 10.0%

---

### Official Review · Reviewer_nWKA · 2024-10-31

**Soundness:** 2
**Presentation:** 2
**Contribution:** 2
**Rating:** 6
**Confidence:** 4

**Summary:**

This paper explores using large models for symbolic music pre-training, proposing ABC Notation as a more compatible format than the commonly used MIDI. To maintain track coherence in multi-track compositions, the authors introduce SMT-ABC Notation. Key contributions include models that handle up to 8192 tokens, covering 90% of their symbolic music dataset, and an examination of the Symbolic Music Scaling Law (SMS Law) on model performance. The experiments show better performance than GPT4.

**Strengths:**

1. Studying a foundation model for symbolic music is both interesting and meaningful work.

2. The team put significant effort into analyzing large datasets and training the foundation model.

3. The findings on scaling laws are interesting.

**Weaknesses:**

1. The research contribution is limited. To enhance the novelty of the approach, consider addressing research questions such as: What factors make ABC better than MIDI, and how does the uniqueness of ABC benefit large models in the feature space? Under what circumstances does the model perform better with ABC notation?

2. The comparison between ABC and MIDI is incomplete. The assumption that ABC is better than MIDI requires a detailed and careful analysis. For example, experiments exploring the following would help clarify the relative strengths and weaknesses of each format for large model training: under the same data size and model architecture, assess whether ABC consistently outperforms MIDI in terms of model efficiency and output quality. Note that MIDI can be represented in various formats, such as piano roll, matrix formats like Compound Word, or sequences like REMI. For Figure 1, further details would improve the visualization, such as comparing ABC directly with MIDI instead of only with the piano roll, as the piano roll omits much information found in MIDI. Indicate to readers what information is missing from both MIDI and ABC, what MIDI can represent (e.g., velocity, control information) that ABC cannot, and vice versa. Additionally, clarify the correspondence between ABC and MIDI, such as how "K: Bb" in ABC notation relates to MIDI, or what "|F2 z G|" represents in ABC and its MIDI equivalent in the figure.

3. The experimental design has flaws. When comparing music-related performance, the proposed model should be evaluated against symbolic music generation models rather than language models. Relevant symbolic music generation baselines include Music Transformer [1], Compound Word Transformer [2], and RWKV [3] on the same large symbolic music dataset.


4. typos such as Line 353.

**Questions:**

Considering the limitations of ABC notation—such as its inability to represent control information and note dynamics, and its less common use within the music community—these weaknesses could significantly impact the model’s potential for generating symbolic music for potential users (composers, musicians, etc). How do you foresee the future roles and applications of ABC and MIDI representations in music generation?

How many songs are included in the dataset, and what detailed information is provided for each song? How do the authors address intellectual property concerns?

**Details Of Ethics Concerns:**

Since the authors use a very large music-related dataset, which might involve intellectual property concerns, it is important to provide a detailed description of the data usage.

---

> ### Author Response · Authors · 2024-11-24
>
> Thank you for your valuable feedback. We acknowledge that our comparison between ABC and MIDI representations is limited, as our **primary focus** is to demonstrate that our model（SMT-ABC-notation-based model）has the potential to **outperform the state-of-the-art symbolic music generation models such as MMT and chatMusician**, as shown in Table 5&6. **An in-depth comparison with ABC notation and other symbolic music formats of MIDI (and more in [1] section)** is not the main contribution or purpose of our paper. Below, we detail our points and why ABC can be a good choice for symbolic music LLMs:
>
>
> ## Advantage of ABC notation
>
> ABC notation offers significant advantages over MIDI in terms of compression ratio, which directly impacts model efficiency and reduces training costs. As highlighted in ChatMusician [4], ABC notation achieves an average of 288.21 tokens per song and 5.16 tokens per second, requiring roughly 38% fewer tokens than various MIDI-based representations.
> This remarkable reduction in token count stems from ABC's ability to efficiently encode symbolic musical repetition using concise notations such as repeat signs (|: and :|). These symbols effectively represent patterns spanning several seconds to minutes. By reducing sequence length, ABC not only lowers computational overhead but also simplifies the learning complexity, making it an ideal format for music-related tasks.
>
> It is undeniable that MIDI offers more detailed dynamic information compared to leadsheet formats like ABC. However, a major drawback of MIDI is that, within a limited context length, it may not be possible to encode an entire song. While hierarchical modeling methods, such as Whole-Song Hierarchical Generation of Symbolic Music Using Cascaded Diffusion Models, have been proposed to address this issue, scaling up such approaches can be challenging.
> In contrast, ABC notation is compatible with natural language symbols and naturally encoded structures like repetition, benefitting training efficiency, convergence speed, and modelling the overall structure of a song with NLP codebase. Moreover, ABC notation integrates seamlessly with text tokenizers, making it an efficient choice. After carefully weighing the pros and cons of both formats and considering the characteristics of our collected dataset, we decided to use ABC notation.
>
> Our proposed SMT-ABC notation effectively addresses the alignment challenges in multitrack ABC notation, significantly improving training loss and overall model performance.
>
> ## Additional Contribution for Scaling up SMT-ABC based LLM
> ### Reserach work on ABC notation remains under-exploration
> As previously discussed, our contribution lies in exploring how to scale up a music generation model and the rationale behind choosing ABC notation, which is naturally similar to natural language. SMT-ABC is designed to ensure that bars played simultaneously across tracks can be inferred causally by the scaled-up model. Music Transformer, RWKV, and Compound Word Transformer are all causal modeling strategies. We believe the SMT-ABC notation dataset can effectively complement these model architectures. We chose LLaMA because it is one of the most widely used causal models. Furthermore, we believe that these models can greatly benefit from our dataset and the SMT-ABC methodology.
>
> ### Scaling Laws and Repetition in ABC Notation Pretraining
> While scaling laws have been extensively studied in other domains, limited research exists on scaling laws specific to ABC notation. Due to the relatively small amount of available ABC-notation data, repetitive training is necessary to achieve meaningful performance. Importantly, ABC notation differs significantly from standard language data, both in structure and representation. Our work addresses this gap by providing insights into scaling laws and repetitive training for ABC data, offering a significant contribution to this underexplored area.
>
> ## Foreesee the future roles and applications
> -	Leadsheet-based multitrack symbolic music generation is a promising yet underexplored area. Leadsheets effectively compress the semantic information of music, facilitating whole-song-level modeling. In the future, this approach could guide audio music generation models similar to Suno's framework and seamlessly integrate text-based leadsheet generation into the framework of large language models (LLMs).
> -	It is also worth noting that audio can be generated from ABC leadsheets by using a rendering model. For example, Seed-Music: A Unified Framework for High Quality and Controlled Music Generation demonstrates this approach, enabling the generation of high-quality audio based on ABC notation. This allows for the inclusion of performance characteristics and timbre while maintaining the capability to model entire songs with ease.

---

> ### Author Response · Authors · 2024-11-24
>
> -	To the best of our knowledge, we have collected the largest ABC notation dataset and successfully scaled model training on this dataset. For the first time, we have derived a scaling law for symbolic music, laying a solid foundation for integrating musical creativity into general artificial intelligence systems in the future.
> -	ABC tokens are similar to the symbols of natural languages, which helps interact with natural languages and can be easily adapted to LLM, such as chatMusician, etc.
>
> ## Typos
> Thank you for pointing out the typos in our manuscript. If the paper is accepted, we will ensure that these issues are corrected in the camera-ready version.
>
> ## Datasets
> Due to the length of the reply, please refer to our response to Reviewer f4e7.
>
> ## The alignment and difference between ABC notation and MIDI
> ### Alignment between 2 notations
> ABC is the music staff/stave in terms of natural language notations, including tune info (e.g. title, composer, meter, key, pitches, rhythms, bar lines & repeats, instrument, etc. [2-3]. The “K: Bb” you mentioned in ABC means the key of the music piece is Bb, which is the same as the key of the MIDI note in the piano roll and is related to the overall pitch of the MIDI note in the piano roll. And “|F2 z G|” corresponds to one music bar that includes a quarter note at pitch F, an eighth-note/quaver rest, and an eighth-note at pitch G. MIDI start and stop times are absolute time slots in seconds and do not include quantized beat/bar information. This given measure corresponds to two MIDI note rectangles on the chart that correspond to the pitch and time (the rest is not shown).
>
> ### The difference between MIDI and ABC
> (1) As you said, the velocity, control information (and also sound effect etc) included in MIDI are not included in ABC notations. ABC notation includes bar and note quantization information (how many beats v.s. how many seconds per note), key, and music/tune structure such as repeat sign, etc., that MIDI does not include. In general, MIDI can keep more information on the styles of performers with acoustic information, and ABC notations are better with modelling repeats and structures, leading to better compression rate and more suitable for LLM training.
>
> [1] Ma Y, Øland A, Ragni A, et al. Foundation models for music: A survey[J]. Foundation models for music: a survey[J]. arXiv preprint arXiv:2408.14340, 2024.
> [2] https://abcnotation.com/wiki/abc:standard:v2.1
> [3] https://abcnotation.com/videos#basics
> [4] Yuan R, Lin H, Wang Y, et al. Chatmusician: Understanding and generating music intrinsically with llm[J]. arXiv preprint arXiv:2402.16153, 2024.

---

### Official Review · Reviewer_FKmb · 2024-11-06

**Soundness:** 3
**Presentation:** 3
**Contribution:** 3
**Rating:** 6
**Confidence:** 2

**Summary:**

The paper presents MuPT a pretrained music transformer for symbolic music generation. The paper is a study of the techniques from LLM literature applied to symbolic music data. The authors discuss a novel data representation for symbolic data: SMT-ABC to take issues related to misaligned multi-track data. The authors also present a scaling law appropriate for symbolic music where the dataset size is typically orders of magnitude smaller than text data that was used to establish the Chinchilla laws. The authors conduct subjective and objective evaluations of their model and compare with other similar models of symbolic music. They find that their model is heavily preferred compared to other models in human studies.

**Strengths:**

The study is conducted in a systematic fashion and covers a lot of the relevant topics that LLM-related literature should talk about, such as the relationship between dataset size, model parameters, etc with the quality of generated data.
The quality of the generated music shared in the supplementary material is decent.

**Weaknesses:**

The authors do not share any samples from the baseline models in supplementary materials which would have helped support the results in the human study.
The repetition metrics seem to not be clearly motivated. Why would the average repetition rate be a meaningful metric. Isn't the more important metric the position of the repeats?
Some writing issues. E.g. missing reference in line 353, missing word in line 505.

**Questions:**

Aside from the questions in the weaknesses section:
- Is there any study to check if the generations are copied from the training data?

---

> ### Author Response · Authors · 2024-11-24
>
> **Samples from baseline model**
>
>
> Thanks for your suggestions， however, we do share the samples in the supplementary materials, which are called MMT-*.mp3. And in this link https://www.notion.so/MuPT-Demos-vs-MIDI-54c19455438f4368a9a43f8ac10a5c01, you can see there are GPT-4 generation examples. According to the subjective participant's results shown in Table 7, Mupt outperforms MMT and GPT-4 in terms of music generation.
>
>
> **The repetition metrics not clearly be motivated**
>
>  We show this in Line 449.
> ```
> Repetition is significant in evaluating how well-structured the music is.
> ```
>
> As mentioned in the paper, we intuitively use the repetition rates to measure how often repetition symbols appear in the generated pieces, as a simple metric for evaluating the structure of music. Theoretically, repetition will largely influence the structureness we perceive about the music. It is a fundamental characteristic of what we experience as music, except for certain types like aleatoric music[1-2]. To the best of our knowledge, some empirical research tries to model the multi-level structure of music by recognising repetition but little of them emphasizes the position of repetition[3-4]. In other words, even a randomly selected note sequence can be regarded as music if it repeats as a pattern. We understand your claim that the position of repetition symbol will influence music quality but it depends on personal preference. Therefore, we do ask the participants to consider the structureness which indicates the repetition and its position by mentioning how well music phrases organize when evaluating musicality in the subjective tests.
>
> [1] Ockelford, A. (2017). Repetition in music: Theoretical and metatheoretical perspectives. Routledge.
> [2] Margulis, E. H. (2013). On repeat: How music plays the mind. Oxford University Press.
> [3] Nieto, O., Mysore, G. J., Wang, C. I., Smith, J. B., Schlüter, J., Grill, T., & McFee, B. (2020). Audio-based music structure analysis: Current trends, open challenges, and applications. Transactions of the International Society for Music Information Retrieval, 3(1).
> [4] Dai, S., Jin, Z., Gomes, C., & Dannenberg, R. B. (2021). Controllable deep melody generation via hierarchical music structure representation. arXiv preprint arXiv:2109.00663.
>
> >Question: Is there any study to check if the generations are copied from the training data?
>
> Yes, we conducted experiments to investigate whether the music generated by the model exhibits any copying from the training data. Specifically, we sampled 750 pieces of music with 1-15 tracks and 100 pieces with more than 15 tracks from the training set. For these samples, we used their headers and the first bar as prompts and calculated the similarity between the model-generated music and the original music from the training set. The similarity was measured using three metrics: n-grams (character-based segmentation), n-gram-music (note/chord-based segmentation), and the longest common subsequence(LCS) between the two music sequences. The results are shown in the table below.
>
> |N-gram(n=4)|N-gram-Music(n=4)|LCS|
> |---|---|---|
> |0.19|0.08|0.29|
>
> The results suggest that the model is capable of generating novel sequences rather than simply copying training data. The low n-gram and n-gram-music scores, paired with the moderate LCS score, indicate that the model retains stylistic or structural influences from the training set without overfitting to exact examples.

---

### Author Response · Authors · 2024-12-04

**Paper title:** MuPT: A Generative Symbolic Music Pretrained Transformer

**Comment:** Dear Reviewers and Chairs,

We sincerely thank all the reviewers again for their insightful and constructive reviews. We are grateful that they found our paper has well-structured (f4e7) and recognized our method to be novel & interesting (Fkmb, nWkA,PkBU), meaningful (nWkA, ku1W) and with high-quality results  (Fkmb,). We explain details and address the concerns of each reviewer and reviewer PkBU has raised his/her score.
Our key contributions are summarized as follows:

- We highlight that leadsheet-based multitrack symbolic music generation is a promising yet underexplored area. Leadsheets effectively compress the semantic information of music, facilitating whole-song-level modeling. We foresee that this approach could guide audio music generation models similar to Suno's framework and seamlessly integrate text-based leadsheet generation into the framework of large language models (LLMs).
- We propose SMT-ABC notation to effectively address the alignment challenges in multitrack ABC notation, significantly improving training loss and overall model performance.
- We have collected the largest ABC notation dataset to date and successfully scaled model training on this dataset. For the first time, we derive a scaling law for symbolic music, laying a solid foundation for integrating musical creativity into general artificial intelligence systems in the future.
- We observe that while scaling laws have been extensively studied in other domains, limited research exists on scaling laws specific to ABC notation. We note that due to the relatively small amount of available ABC-notation data, repetitive training is necessary to achieve meaningful performance. Importantly, we address the unique challenges posed by the significant differences between ABC notation and standard language data, both in structure and representation. Through our work, we provide insights into scaling laws and repetitive training for ABC data, offering a significant contribution to this underexplored area.
- We have also revised some minor details suggested by reviewers and provided implementation details for the new experiments added.

Thank you all once again for your valuable feedback, dedication, engagement and attention; we greatly appreciate them.

Best Regards,

Authors

---

### Meta-Review · Area_Chair_qVN6 · 2024-12-20

**Metareview:**

The paper presents a modification to ABC notation that interleaves different tracks in time based on a bar alignment. This representation was used to train an LLM model. Results were compared with other multi-track models and human evaluation. Scaling analysis of loss as a function of the number of tokens was conducted, suggesting an optimal model parameter size.

**Additional Comments On Reviewer Discussion:**

Although all comments of the reviewers were positive and ranked above threshold, my main concern is that the research contribution is limited to a representation change, while relying on existing LLM without considering functional or structural aspects or any modifications to the training or prompting procedures. The comment of reviewer nWKA about flaws in experimental design, asking for including baselines of Compound Word Transformer, and RWKV on the same large symbolic music dataset, was not experimentally pursued.

---

### Decision · Program_Chairs · 2025-01-22

Accept (Poster)